# Diminishing Return of Value Expansion Methods in Model-Based Reinforcement Learning

**Daniel Palenicek** [1 2]    **Michael Lutter** [1]    **João Carvalho** [1]    **Jan Peters** [1 2 3 4]

## Abstract

Model-based reinforcement learning is one approach to increase sample efficiency. However, the accuracy of the dynamics model and the resulting compounding error over modelled trajectories are commonly regarded as key limitations. A natural question to ask is: How much more sample efficiency can be gained by improving the learned dynamics models? Our paper empirically answers this question for the class of model-based value expansion methods in continuous control problems. Value expansion methods should benefit from increased model accuracy by enabling longer rollout horizons and better value function approximations. Our empirical study, which leverages oracle dynamics models to avoid compounding model errors, shows that (1) longer horizons increase sample efficiency, but the gain in improvement decreases with each additional expansion step, and (2) the increased model accuracy only marginally increases the sample efficiency compared to learned models with identical horizons. Therefore, longer horizons and increased model accuracy yield diminishing returns in terms of sample efficiency. These improvements in sample efficiency are particularly disappointing when compared to model-free value expansion methods. Even though they introduce no computational overhead, we find their performance to be on-par with model-based value expansion methods. Therefore, we conclude that the limitation of model-based value expansion methods is not the model accuracy of the learned models. While higher model accuracy is beneficial, our experiments show that even a perfect model will not provide an un-rivalled sample efficiency but that the bottleneck lies elsewhere.

## 1 Introduction

Insufficient sample efficiency is a central issue that prevents reinforcement learning (RL) agents from learning in physical environments. Especially in applications like robotics, samples from the real system are particularly scarce and expensive to acquire due to the high cost of operating robots. A technique that has proven to substantially enhance sample efficiency is model-based reinforcement learning (Deisenroth et al., 2013). In model-based RL, a model of the system dynamics is usually learned from data, which is subsequently used for planning (Chua et al., 2018; Hafner et al., 2019) or for policy learning (Sutton, 1990; Janner et al., 2019).

Over the years, the model-based RL community has identified several ways of applying (learned) dynamics models in the RL framework. Sutton's (1990) DynaQ framework uses the model for data augmentation, where model rollouts are started from real environment states coming from a replay buffer. The collected data is afterwards given to a model-free RL agent. Recently, various improvements have been proposed to the original DynaQ algorithm, such as using ensemble neural network models and short rollout horizons (Janner et al., 2019; Lai et al., 2020), and improving the synthetic data generation with model predictive control (Morgan et al., 2021).

Feinberg et al. (2018) proposed an alternative way of incorporating a dynamics model into the RL framework. Their model-based value expansion (MVE) algorithm unrolls the dynamics model and

---

[1] Intelligent Autonomous Systems, Technical University of Darmstadt, daniel.palenicek@tu-darmstadt.de.

[2] Hessian.AI, Hochschulstr. 10, 64293 Darmstadt, Germany.

[3] German Research Center for AI (DFKI), Research Department: Systems AI for Robot Learning.

[4] Centre for Cognitive Science, Hochschulstr. 10, 64293 Darmstadt, Germany.

discounts along the modelled trajectory to approximate better targets for value function learning. Subsequent works were primarily concerned with adaptively setting the rollout horizon based on some (indirect) measure of the modelling error, e.g., model uncertainty (Buckman et al., 2018; Abbas et al., 2020), a reconstruction loss (Wang et al., 2020) or approximating the local model error through temporal difference learning (Xiao et al., 2019).

Lastly, using backpropagation through time (BPTT) (Werbos, 1990), dynamics models have been applied to the policy improvement step to assist in computing better policy gradients. Deisenroth & Rasmussen (2011) use Gaussian process regression (Rasmussen & Williams, 2006) and moment matching to find a closed-form solution for the gradient of the trajectory loss function. Stochastic value gradients (SVG) (Heess et al., 2015) uses a model in combination with the reparametrization trick (Kingma & Welling, 2014) to propagate gradients along real environment trajectories. Others leverage the model's differentiability to directly differentiate through model trajectories (Byravan et al., 2020; Amos et al., 2021). On a more abstract level, MVE- and SVG-type algorithms are very similar. Both learn a dynamics model and use it for $H$-step trajectories (where $H$ is the number of modelled time steps) to better approximate the quantity of interest – the next state $Q$-function in the case of SVG and the $Q$-function for MVE. These value expansion methods all assume that longer rollout horizons will improve learning if the model is sufficiently accurate.

The common opinion is that learning models with less prediction error may further improve the sample efficiency of RL. This argument is based on the fact that the single-step approximation error can become a substantial problem when using learned dynamics models for long trajectory rollouts. Minor modelling errors can accumulate quickly when multi-step trajectories are built by bootstrapping successive model predictions. This problem is known as *compounding model error*. Furthermore, most bounds in model-based RL are usually dependent on the model error (Feinberg et al., 2018), and improvement guarantees assume model errors converging to zero. In practice, rollout horizons are often kept short to avoid significant compounding model error build-up (Janner et al., 2019). Intuitively, using longer model horizons has the potential to exploit the benefits of the model even more. For this reason, immense research efforts have been put into building and learning better dynamics models. We can differentiate between purely engineered (white-box) models and learned (black-box) models (Nguyen-Tuong & Peters, 2011). White-box models offer many advantages over black-box models, as they are more interpretable and, thus, more predictable. However, most real-world robotics problems are too complex to model analytically, and one has to retreat to learning (black-box) dynamics models (Chua et al., 2018; Janner et al., 2020). Recently, authors have proposed neural network models that use physics-based inductive biases, also known as grey-box models (Lutter et al., 2019a;b; Greydanus et al., 2019).

While research has focused on improving model quality, a question that has received little attention yet is: ***Is model-based reinforcement learning even limited by the model quality?*** For Dyna-style data augmentation algorithms, the answer is yes, as these methods treat model- and real-environment samples the same. For value expansion methods, i.e., MVE- and SVG-type algorithms, the answer is unclear. Better models would enable longer horizons due to the reduced compounding model error and improve the value function approximations. However, the impact of both on the sample efficiency remains unclear. In this paper, we empirically address this question for value expansion methods. Using the true dynamics model, we empirically show that the sample efficiency does not increase significantly even when a perfect model is used. We find that increasing the rollout horizon with oracle dynamics as well as improving the value function approximation using an oracle model compared to a learned model at the same rollout horizon yields *diminishing returns* in improving sample efficiency. With the phrase *diminishing returns*, we refer to its definition in economics (Case & Fair, 1999). In the context of value expansion methods in model-based RL, we mean that the marginal utility of better models on sample efficiency significantly decreases as the models improve in accuracy. These gains in sample efficiency of model-based value expansion are especially disappointing compared to model-free value expansion methods, e.g., Retrace (Munos et al., 2016). While the model-free methods introduce no computational overhead, the performance is on-par compared to model-based value expansion methods. When comparing the sample efficiency of different horizons for both model-based and model-free value expansion, one can clearly see that improvement in sample efficiency at best decreases with each additional model step along a modelled trajectory. Sometimes the overall performance even decreases for longer horizons in some environments.

**Contributions.** In this paper, we empirically address the question of whether the sample efficiency of value expansion methods is limited by the dynamics model quality. We summarize our contributions as follows. (1) Using the oracle dynamics model, we empirically show that the sample efficiency of value expansion methods does not increase significantly over a learned model. (2) We find that increasing the rollout horizon yields diminishing returns in improving sample efficiency, even with oracle dynamics. (3) We show that off-policy Retrace is a very strong baseline, which adds virtually no computational overhead. (4) Our experiments show that for the critic expansion, the benefits of using a learned model does not appear to justify the added computational complexity.

## 2  $H$-STEP POLICY EVALUATION AND IMPROVEMENT

In this section, we introduce the necessary background and components of value expansion.

**Reinforcement Learning.** Consider a discrete-time Markov Decision Process (MDP) (Puterman, 2014), defined by the tuple $\langle \mathcal{S}, \mathcal{A}, \mathcal{P}, \mathcal{R}, \rho, \gamma \rangle$ with state space $\mathcal{S}$, action space $\mathcal{A}$, transition probability $s_{t+1} \sim \mathcal{P}(\cdot | s_t, a_t)$, reward function $r_t = \mathcal{R}(s_t, a_t)$, initial state distribution $s_0 \sim \rho$ and discount factor $\gamma \in [0, 1)$. At each time step $t$ the agent interacts with the environment according to its policy $\pi$. Wherever it is clear, we will make use of the tick notation $s'$ and $a'$ to refer to the next state and action at time $t + 1$. A trajectory $\tau = (s_0, a_0, s_1, a_1, \ldots, s_T)$ is a sequence of states and actions. The objective of an RL agent is to find an optimal policy that maximizes the expected sum of discounted rewards $\pi^* = \arg\max_\pi \mathbb{E}_{\tau \sim \rho, \pi, \mathcal{P}} \left[ \sum_{t=0}^\infty \gamma^t r_t \right]$. Often $\pi$ is parametrized and, in the model-free case, it is common to optimize its parameters with on-policy (Sutton et al., 1999; Peters & Schaal, 2008; Peters et al., 2010; Schulman et al., 2015; 2017; Palenicek, 2021) or off-policy gradient methods (Degris et al., 2012; Silver et al., 2014; Lillicrap et al., 2016; Fujimoto et al., 2018; Haarnoja et al., 2018). In model-based RL an approximate model $\hat{\mathcal{P}}$ of the true dynamics is learned, and optionally an approximate reward function $\hat{\mathcal{R}}$. This model is then used for data generation (Sutton, 1990; Janner et al., 2019; Cowen-Rivers et al., 2022), planning (Chua et al., 2018; Hafner et al., 2019; Lutter et al., 2021a;b; Schneider et al., 2022) or stochastic optimization (Deisenroth & Rasmussen, 2011; Heess et al., 2015; Clavera et al., 2020; Amos et al., 2021).

**Maximum-Entropy Reinforcement Learning** (MaxEnt RL) augments the MDP's reward $r_t$ with a policy entropy term, which prevents collapsing to a deterministic policy, thus implicitly enforcing continuing exploration (Ziebart et al., 2008; Fox et al., 2016; Haarnoja et al., 2017). The (soft) action-value function $Q^\pi(s, a) = \mathbb{E}_{\pi, \mathcal{P}} \left[ \sum_{t=0}^\infty \gamma^t (r_t - \alpha \log \pi(a_t | s_t)) \mid s_0 = s, a_0 = a \right]$ computes the expected sum of discounted soft-rewards following a policy $\pi$. The goal is to maximize $\mathcal{J}_\pi = \mathbb{E}_{s_0 \sim \rho} \left[ \sum_{t=0}^\infty \gamma^t (r_t - \alpha \log \pi(a_t | s_t)) \right]$ w.r.t. $\pi$. Haarnoja et al. (2018) have proposed the Soft Actor-Critic (SAC) algorithm to solve this objective. The SAC actor and critic losses are defined as

$$\mathcal{J}_\pi(\mathcal{D}) = \mathbb{E}_{s \sim \mathcal{D}} \left[ -V^\pi(s) \right] = \mathbb{E}_{s \sim \mathcal{D}, \, a \sim \pi(\cdot|s)} \left[ \alpha \log \pi(a|s) - Q^\pi(s, a) \right], \tag{1}$$

$$\mathcal{J}_Q(\mathcal{D}) = \mathbb{E}_{(s,a,r,s') \sim \mathcal{D}} \left[ \left( r + \gamma V^\pi(s') - Q^\pi(s, a) \right)^2 \right], \tag{2}$$

where $\mathcal{D}$ is a dataset of previously collected transitions (replay buffer), and the next state value target is $V^\pi(s') = \mathbb{E}_{a' \sim \pi(\cdot|s')} \left[ Q^\pi(s', a') - \alpha \log \pi(a'|s') \right]$. Note, that $r_t + \gamma V^\pi(s')$ is a target value. Thus, we do not differentiate through it, as is common in deep RL (Riedmiller, 2005; Mnih et al., 2013).

**Value Expansion.** Through the recursive definition of the value- and action-value functions (Bellman, 1957), they can be approximated by rolling out the model for $H$ time steps. We refer to this (single sample) estimator as the $H$-step value expansion, defined as

$$Q^H(s, a) = \mathcal{R}(s, a) + \sum_{t=1}^{H-1} \gamma^t \left( r_t - \alpha \log \pi(a_t | s_t) \right) + \gamma^H V^\pi(s_H) \tag{3}$$

with $Q^0(s, a) := Q^\pi(s, a)$ and $V^\pi(s) = \mathbb{E}_{a \sim \pi(\cdot|s)} \left[ Q^\pi(s, a) - \alpha \log \pi(a|s) \right]$. For $H = 0$ this estimator reduces to the function approximation used in SAC. It is important to note that these trajectories are on-policy. In practice, the SAC implementation uses a double $Q$-function (Fujimoto et al., 2018; Haarnoja et al., 2018), but to keep the notation simple we do not include it here. We further evaluate the expectation for the next state value function with a single sample, i.e. $V^\pi(s') = Q^\pi(s', a') - \log \pi(a'|s')$ with $a' \sim \pi(\cdot \mid s')$, as done in Haarnoja et al. (2018).

**Actor Expansion.** The $H$-step value expansion can now be used in the actor update by incorporating Equation 3 in the actor loss (Equation 1)

$$\mathcal{J}_\pi^H(\mathcal{D}) = \mathbb{E}_{s\sim\mathcal{D},\,a\sim\pi(\cdot|s)}\big[\alpha\log\pi(a|s) - Q^H(s,a)\big], \tag{4}$$

which resembles the class of SVG algorithms (Heess et al., 2015; Byravan et al., 2020; Amos et al., 2021; Clavera et al., 2020). The difference to the original SVG formulation is the addition of the entropy term. Note that, for $H = 0$ this definition reduces to the regular SAC actor loss (Equation 1). We will refer to the $H$-step value expansion for the actor update as *actor expansion* (AE).

**Critic Expansion.** Incorporating value expansion in the policy evaluation step results in TD-$\lambda$ style (Sutton & Barto, 2018; Schulman et al., 2016) or Model-Based Value Expansion (MVE) methods (Wang et al., 2020; Buckman et al., 2018). The corresponding critic loss is defined as

$$\mathcal{J}_Q^H(\mathcal{D}) = \mathbb{E}_{(s,a,r,s')\sim\mathcal{D},\,a'\sim\pi(\cdot|s')}\left[\big(r + \gamma(Q^H(s',a') - \alpha\log\pi(a'|s')) - Q^\pi(s,a)\big)^2\right]. \tag{5}$$

Similar to actor expansion, for $H = 0$ the critic update reduces to the SAC critic update (Equation 2). We will refer to the $H$-step value expansion for the critic update as *critic expansion* (CE).

**Extension to Off-Policy Trajectories.** We can naturally extend the action-value expansion in Equation 3 with the Retrace formulation (Munos et al., 2016), which allows to use off-policy trajectories collected with a policy $\mu \neq \pi$ (e.g. stored in a replay buffer). The $H$-step Retrace state-action value expansion (single sample) estimator is defined as

$$Q_{\text{retrace}}^H(s,a) = \mathcal{R}(s,a) + \left[\sum_{t=1}^{H-1}\gamma^t\left(c_t r_t + c_{t-1}V(s_t) - c_t Q(s_t,a_t)\right)\right] + \gamma^H c_{H-1}V(s_H), \tag{6}$$

with the importance sampling weight $c_t = \prod_{j=1}^t \lambda\min\left(1, \pi(a_j|s_j)/\mu(a_j|s_j)\right)$ and $c_0 := 1$. We set $\lambda = 1$ for the remainder of this paper. And by definition $Q_{\text{retrace}}^0(s,a) := Q^\pi(s,a)$. The Retrace extension has the desirable property that $c_t = 1$ for on-policy trajectories (i.e. $\mu = \pi$), and in that case $Q_{\text{retrace}}^H$ reduces to the $H$-step action-value target $Q^H$ (Equation 3, see Appendix A.2 for a detailed derivation). Furthermore, for $H = 0$, it reduces to the SAC target. The second desirable property is that trajectories from the real environment or a dynamics model can be used. Hence, we use the Retrace formulation to evaluate all combinations of CE and AE with on- and off-policy data and real and model-generated data. Since Retrace encompasses all the variants we introduced, for readability, we will from now on overload notation and define $Q^H(s,a) := Q_{\text{retrace}}^H(s,a)$.

A natural combination is to use actor and critic expansion simultaneously. However, we analyze each method separately to better understand their issues. Table A.1 provides an overview of related value expansion literature and displays further information about their usage of actor and critic expansion and on- and off-policy data.

## 3 EMPIRICAL STUDY

In this experimental section, we investigate the initial research question, **how much will more accurate dynamics models increase sample efficiency?** In our experimental setup we replace the learned dynamics model with an oracle model – the environment's simulator – which is the most accurate model that could be learned. In this scenario, no model error exists, which lets us examine the impact of the compounding model error — or rather, the lack of it. In these experiments, we specifically exclude data augmentation methods, where it is trivial to realize that they can arbitrarily benefit from a perfect dynamics model. We conduct a series of empirical studies across the introduced value expansion algorithms on five standard continuous control benchmark tasks: InvertedPendulum, Cartpole SwingUp, Hopper, Walker2d, and Halfcheetah.[1] These range from simple to complex, with the latter three including contact forces, often resulting in discontinuous dynamics.

In the following subsections, we first present the premise of this paper — the two types of diminishing returns of value expansion methods (Section 3.1). Given the oracle model, we *rule out the existence of a compounding model error* and, therefore, possible impacts on the learning performance. Next, we turn to the generated $H$-step target values. We show that the increasing target

---

[1]The code is available at: https://github.com/danielpalen/value_expansion.

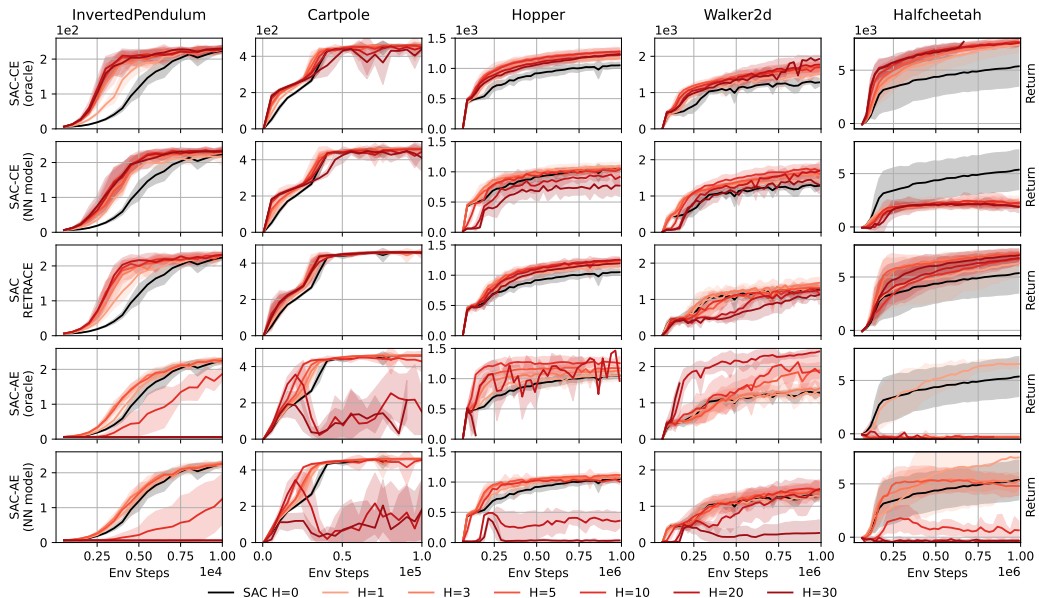

Figure 1: Shows the diminishing return of critic expansion and actor expansion methods using multiple rollout horizons $H$ for a SAC agent. CE does not benefit noticeably from horizons larger than 5 in most cases. There is no clear trend that CE methods benefit from better models since the neural network model performs mostly on par with the oracle model. Model-free Retrace performances comparably. For AE (rows 4 & 5), larger horizons can be detrimental, even with an oracle model. We plot the average episode undiscounted return mean (solid line) and standard deviation (shaded area) against the number of real environment interaction steps for 9 random seeds. Some AE runs for larger $H$ terminate early due to exploding gradients (See Section 3.3).

variance caused by longer rollout horizons is not linked to the diminishing returns (Section 3.2). Finally, in Section 3.3, we look into the critic and actor gradients. We implement the learned model as a stochastic neural network ensemble, similar to PETS and MBPO (See Appendix A.3 for details).

## 3.1 DIMINISHING RETURNS ON CONTINUOUS CONTROL TASKS

Figure 1 shows the learning curves of the different value expansion methods for multiple rollout horizons and model types based on a SAC agent. Each model-based value expansion method, i.e., critic expansion (CE) and actor expansion (AE) is shown with an oracle and a learned model. The rollout horizons range from very short to long $H \in [0, 1, 3, 5, 10, 20, 30]$, where $H = 0$ reduces to vanilla SAC. Larger horizons $H \gg 30$ are not considered as the computational time complexity becomes unfeasible (see Table A.3). In addition to comparing model-based value expansion methods, Figure 1 also shows learning curves of the model-free value expansion method Retrace.

**Diminishing Return of Rollout Horizons.** Within short horizons, less than 5 steps, we consistently observe that a single increase in the rollout horizon leads to increased performance in most cases. However, the rate of improvement quickly decreases with each additional expansion step (see Figure 1). Close to all variants of SAC reach their peak performance for $H = 5$. For more than 5 steps, the sample efficiency and obtained reward do not increase anymore and can even decrease. We refer to this phenomenon as the *diminishing return* of rollout horizon. SAC-CE (1st & 2nd rows) performs better than SAC-AE (4th & 5th rows) for both oracle and learned dynamics models. While for shorter horizons, the performance is comparable, the performance of SAC-AE degrades significantly for longer horizons. For SAC-AE, learning becomes unstable, and no good policy can be learned for most systems using longer horizons. Thus, increasing rollout horizons does not yield unrivalled sample efficiency gains. As diminishing returns w.r.t. rollout horizon are observed for the oracle and learned dynamics model, the reduced performance of the longer rollout horizon cannot be explained by the compounding model errors as hypothesised by prior work.

**Diminishing Return of Model Accuracy.** When directly comparing the performance of the learned and the oracle dynamics models, the learning curves are very similar. The learning performance of the learned model is only slightly reduced compared to the oracle model, which is expected. However, the degraded performance is barely visible for most systems when comparing the learning curves. This trend can be observed across most environments, algorithms and horizons. Therefore, the reduced model accuracy and the subsequent compounding model error do not appear to significantly impact the sample efficiency of the presented value expansion methods. We conclude that one cannot expect large gains in sample efficiency with value expansion methods by further investing in developing more accurate dynamics models. This is not to say that poor dynamics models do not harm the learning performance of value expansion methods. Rather, we observe that overcoming the small errors of current models towards oracle dynamics will, at best, result in small improvements for value expansion methods on the continuous control tasks studied.

**Model-free Value Expansion.** When comparing the model-based value expansion methods to the model-free value expansion methods (Figure 1 3rd row), the sample efficiency of the model-based variant is only marginally better. Similar to the model-based variants, the model-free Retrace algorithm increases the sample efficiency for shorter horizons but decreases in performance for longer horizons. This minor difference between both approaches becomes especially surprising as the model-based value expansion methods introduce a large computational overhead. While the model-free variant only utilizes the collected experience, the model-based variant requires learning a parametric model and unrolling this model at each update of the critic and the actor. Empirically the model-free variant is up to $15\times$ faster in wall-clock time compared to the model-based value expansion at $H = 5$. Therefore, model-free value expansion methods are a very strong baseline to model-based value expansion methods, especially when considering the computational overhead.

## 3.2 INCREASING VARIANCE OF $Q^H(s, a)$ DOES NOT EXPLAIN DIMINISHING RETURNS

One explanation for the diminishing returns could be the increasing variance of the $Q$-function targets as SAC estimates $Q^H(s, a)$ by rolling out a model with a stochastic policy. This unrolling can be seen as traversing a Markov reward process for $H$ steps, for which it is well known that the target estimator's variance increases linearly with $H$, similar to the REINFORCE policy gradient algorithm (Pflug, 1996; Carvalho et al., 2021). We also observed the increase in variance of the $Q$-function targets empirically when increasing the horizon. The experiments highlighting the increase in variance are described in Appendix A.4. To test whether the increased variance of the value expansion targets explains the diminishing returns, we conduct the following two experiments. First, we perform the same experiments as in Section 3.1 with DDPG. As DDPG uses a deterministic policy and the oracle dynamics are deterministic, increasing the horizon does not increase the variance of the $Q$-function targets. Second, we use variance reduction when computing the targets with SAC by averaging over multiple particles per target.

**Deterministic Policies.** Figure 2 shows the learning curves of DDPG for the same systems and horizons as Figure 1. Since both the oracle dynamics and the policy are deterministic, increasing the horizon does not increase the variance of the $Q$-function targets. DDPG shows the same diminishing returns as SAC. When increasing the horizon, the gain in sample efficiency decreases with each additional step. Longer horizons are not consistently better than shorter horizons. For longer horizons, the obtained reward frequently collapses, similar to SAC. Therefore, the increasing variance in the $Q$-function targets cannot explain the diminishing returns.

**Variance Reduction through Particles.** To reduce the $Q$-function target variance when using SAC, we average the computed target $Q^H(s, a)$ over multiple particles per target estimation. For SAC-CE, we used 30 particles and 10 particles for SAC-AE. The corresponding learning curves are shown in Figure 3. Comparing the experiments with variance reduction with the prior results from Figure 1, there is no qualitative difference in training performance. Increasing the horizons by one additional step does not necessarily yield better sample efficiency. As in the previous experiments, for longer horizons, the learning curves frequently collapse, and the performance deteriorates even with the oracle dynamics model and averaging over multiple particles. Therefore, the increasing variance of the SAC $Q$-function targets does not explain the diminishing returns of value expansion methods.

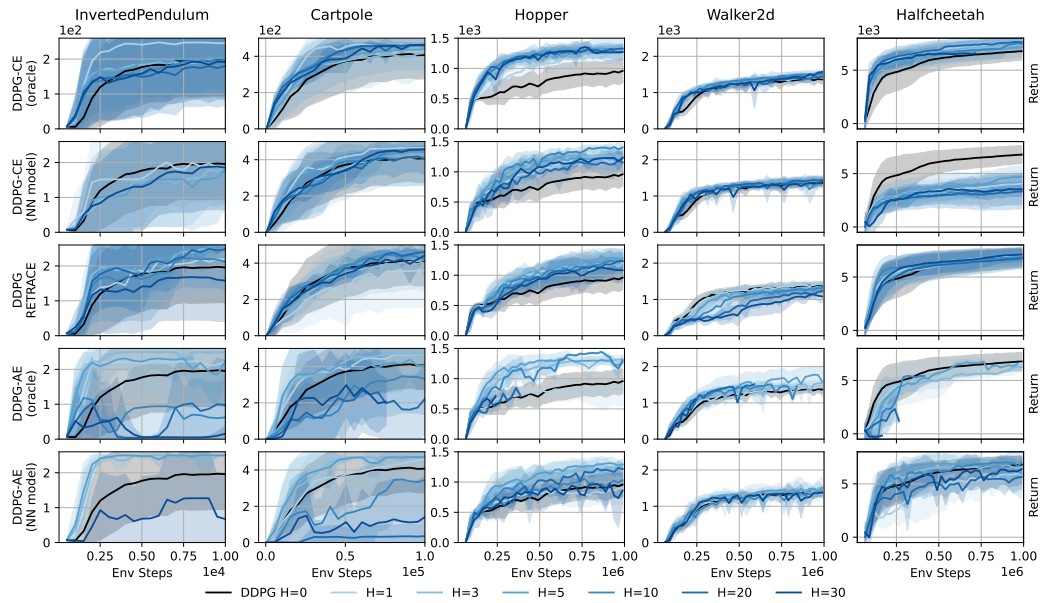

Figure 2: Similar to Figure 1, this plot shows the diminishing return of critic expansion and actor expansion methods using multiple rollout horizons $H$ but for a DDPG agent. We notice the same behaviours, where CE does not benefit noticeably from larger horizons in most cases. There is also no clear trend that CE methods benefit from better models since the neural network model performs mostly on par with the oracle model. Again, model-free Retrace performance is comparable. Rows 4 & 5 show that for AE, larger horizons can be detrimental for DDPG as well, even with an oracle model. We plot the average episode undiscounted return mean (solid line) and standard deviation (shaded area) against the number of real environment interaction steps for 9 random seeds. Some AE runs for larger $H$ terminate early due to exploding gradients (See Section 3.3).

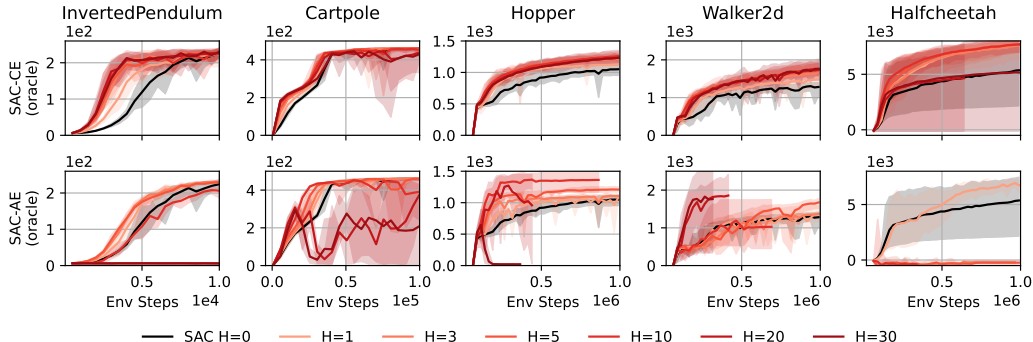

Figure 3: Variance reduction through particles for SAC-CE and SAC-AE experiments. Each $Q$-function target is averaged over multiple particles to reduce the variance (30 particles for SAC-CE and 10 for SAC-AE). Compared to experiments from Figure 1, we see that variance reduction gives the same qualitative training performance. Therefore, the increased variance of the $Q$-function targets is not the root cause of the diminishing returns in value expansion.

## 3.3 IMPACT ON THE GRADIENTS

Lastly, we analyze the gradient statistics during training to observe whether the gradient deteriorates for longer horizons. The gradient variance for a subset of environments and different horizons is shown in Figure 4. In the Appendix, the gradient mean and variance is shown for all environments,

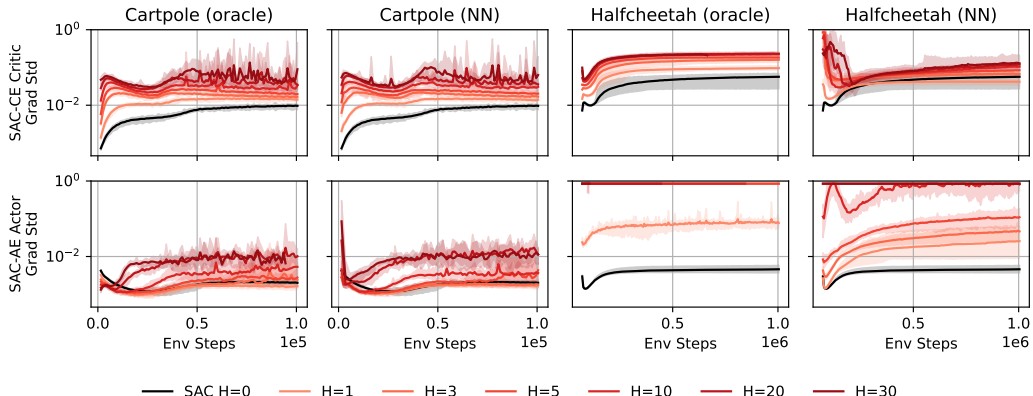

Figure 4: Critic and actor gradient's standard deviation for Cartpole and Halfcheetah and different rollout horizons with both an oracle dynamics model and a learned dynamics model. (a) We observe that the SAC-CE critic gradients show low variance. (b) We see that the SAC-AE actor gradient variance explodes for Halfcheetah with longer horizons. In this Figure, we clip values to $10^0$, which becomes necessary for visualizing longer horizons on the Halfcheetah due to exploding gradients.

including both SAC and DDPG. See Figure A.3 & A.4 for all SAC experiments and Figure A.5 & A.6 for the DDPG experiments.

**Critic Expansion Gradients.** The first row of Figure 4 shows the SAC-CE critic gradient's standard deviation (averaged over a replay buffer batch), over all dimensions of the $Q$-function parameter space, along the training procedure of SAC-CE. For all environments and horizons, the gradient mean and standard deviation appear within reasonable orders of magnitude (also compare Figure A.3 in the Appendix). Although, as expected, we observe a slight increase in the standard deviation with an increasing horizon for both the oracle and the learned neural network model. The mean is around zero ($10^{-5}$) and the standard deviation between $10^{-3}$ and $10^{-1}$. Therefore, for both SAC-CE and DDPG-CE there is no clear signal in the gradient's mean or standard deviation that links to the diminishing returns of CE methods. The little difference w.r.t. horizon length is expected because the critic gradient $\nabla_Q \mathcal{J}_Q^H$ of the loss (Equation 5) is only differentiating through $Q^\pi$ in the squared loss since $Q^H(s', a')$ is assumed to be a fixed target, which is a common assumption in most RL algorithms (Ernst et al., 2005; Riedmiller, 2005; Mnih et al., 2013).

**Actor Expansion Gradients.** The second row of Figure 4 shows the actor gradient's standard deviation (averaged over a replay buffer batch), over all dimensions of the policy parameter space, along the training process for SAC-AE. Contrary to CE, AE backpropagates through the whole rollout trajectory to compute the gradient w.r.t. the policy parameters. This type of recursive computational graph is well known to be susceptible to vanishing or exploding gradients (Pascanu et al., 2013). Comparing rows four and five of Figure 1, we observe that a better dynamics model does not directly translate into faster learning or higher rewards since both the oracle and neural network model show the same effect – lower returns for larger horizons. For SAC-AE we partially correlate these results with the actor gradients in Figure 4. For example, the standard deviation of the Halfcheetah actor gradients explodes for horizons larger or equal to 3 for the oracle and 20 for the learned models. This problem of exploding gradients can also be seen in the Hopper environment using learned dynamics (Appendix Figure A.4). This increase in variance can be correlated with the lower returns as SAC-AE frequently fails to solve the tasks for long horizons. A similar phenomenon can be seen in the DDPG-AE experiments as well, where the policy and oracle dynamics are deterministic. Therefore, increasing the horizon must be done with care even if a better model does not suffer from compounding model errors. For AE, building a very accurate differentiable model that predicts into the future might not be worth it due to gradient instability. For AE, it might only be possible to expand for a few steps before requiring additional methods to handle gradient variance (Parmas et al., 2018).

## 4 CONCLUSION & FUTURE WORK

In this paper, we answer the question of whether dynamics model errors are the main limitation for improving the sample efficiency of model-based value expansion methods. Our evaluations focus on value expansion methods as for model-based data augmentation (i.e. Dyna-style RL algorithms) approaches clearly benefit from a perfect model. By using an oracle dynamics model and thus avoiding model errors altogether, our experiments show that for infinite horizon, continuous state- and action space problems with deterministic dynamics, the benefits of better dynamics models for value-expansion methods yield diminishing returns. More specifically, we observe the *diminishing return of model accuracy* and *diminishing return of rollout horizon*. (1) More accurate dynamics models do not significantly increase the sample efficiency of model-based value expansion methods, as the sample efficiency of the oracle dynamics model and learned dynamics models are comparable. (2) Increasing the rollout horizons by an additional step yields decreasing gains in sample efficiency at best. The gains in sample efficiency quickly plateau for longer horizons, even for oracle models, without compounding model error. For some environments, longer horizons even hurt sample efficiency. Furthermore, we observe that model-free value expansion delivers on-par performance compared to model-based methods without adding any additional computational overhead. Therefore making the model-free value expansion methods a very strong baseline. Overall, our results suggest that the computational overhead of model-based value expansion often outweighs the benefits of model-based value expansion methods, compared to model-free value expansion methods, especially in cases where computational resources are limited.

ACKNOWLEDGMENTS

This work was funded by the Hessian Ministry of Science and the Arts (HMWK) through the projects "The Third Wave of Artificial Intelligence - 3AI" and Hessian.AI. Calculations for this research were conducted on the Lichtenberg high-performance computer of the TU Darmstadt. João Carvalho is funded by the German Federal Ministry of Education and Research (project IKIDA, 01IS20045).

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

# A    APPENDIX

## A.1    RELATED WORK

Table A.1: Overview of key related literature. We categorize model-based RL algorithms using value expansion in the actor and/or critic update, and further note which ones use on-policy trajectories resulting from a dynamics model and which use off-policy trajectories from a replay buffer.

| Algorithm | Expansion | | Policy | |
|---|---|---|---|---|
| | Actor | Critic | On | Off |
| SVG (Heess et al., 2015) | $\times$ | | | $\times$ |
| IVG (Byravan et al., 2020) | $\times$ | | $\times$ | |
| SAC-SVG (Amos et al., 2021) | $\times$ | | $\times$ | |
| MAAC (Clavera et al., 2020) | $\times$ | | $\times$ | |
| MVE (Feinberg et al., 2018) | | $\times$ | $\times$ | |
| STEVE (Buckman et al., 2018) | | $\times$ | $\times$ | |
| Retrace (Munos et al., 2016) | | $\times$ | | $\times$ |
| Dreamer (Hafner et al., 2020) | $\times$ | $\times$ | $\times$ | |

## A.2    FROM RETRACE TO ON-POLICY $H$-STEP VALUE EXPANSION

In this section, we show how Retrace reduces to the $H$-step value expansion target (Equation 3) in the on-policy case, with $\lambda = 1$. First, we perform a simple rewrite from the original Retrace formulation in Munos et al. (2016) such that it is more in line with the notation in this paper. Note that in Retrace, the rollouts are generated with a policy $\mu$. While the value function $V^\pi$ corresponds to the current policy $\pi$.

$$Q_{\text{retrace}}^H(s_0, a_0) = Q(s_0, a_0) + \mathbb{E}_{a_t \sim \mu}\left[\sum_{t=0}^H \gamma^t c_t(\mathcal{R}(s_t, a_t) + \gamma V^\pi(s_{t+1}) - Q(s_t, a_t))\middle| s_0,\ a_0\right]$$

$$= Q(s_0, a_0) + \mathbb{E}_{a_t \sim \mu}\left[\left[\sum_{t=0}^H \gamma^t c_t(\mathcal{R}(s_t, a_t) - Q(s_t, a_t))\right] + \left[\sum_{t=0}^H \gamma^{t+1} c_t V^\pi(s_{t+1})\right]\middle| s_0,\ a_0\right]$$

$$= Q(s_0, a_0) + \mathbb{E}_{a_t \sim \mu}\left[\left[\sum_{t=0}^H \gamma^t c_t(\mathcal{R}(s_t, a_t) - Q(s_t, a_t))\right] + \left[\sum_{t=1}^{H+1} \gamma^t c_{t-1} V^\pi(s_t)\right]\middle| s_0,\ a_0\right]$$

$$= Q(s_0, a_0) + \mathbb{E}_{a_t \sim \mu}\left[\gamma^0 c_0(\mathcal{R}(s_0, a_0) - Q(s_0, a_0))\right]$$

$$+ \mathbb{E}_{a_t \sim \mu}\left[\left[\sum_{t=1}^H \gamma^t c_t(\mathcal{R}(s_t, a_t) - Q(s_t, a_t))\right] + \left[\sum_{t=1}^H \gamma^t c_{t-1} V^\pi(s_t)\right]\right]$$

$$+ \mathbb{E}_{a_t \sim \mu}\left[\gamma^{H+1} c_H V^\pi(s_{H+1})\right]$$

$$= Q(s_0, a_0) + \mathcal{R}(s_0, a_0) - Q(s_0, a_0)$$

$$+ \mathbb{E}_{a_t \sim \mu}\left[\sum_{t=1}^H \gamma^t c_t(\mathcal{R}(s_t, a_t) - Q(s_t, a_t)) + \gamma^t c_{t-1} V^\pi(s_t) + \gamma^{H+1} c_H V^\pi(s_{H+1})\right]$$

$$= \mathcal{R}(s_0, a_0) + \mathbb{E}_{a_t \sim \mu}\left[\sum_{t=1}^H \gamma^t (c_t \mathcal{R}(s_t, a_t) + c_{t-1} V^\pi(s_t) - c_t Q(s_t, a_t)) + \gamma^{H+1} c_H V^\pi(s_{H+1})\right]$$

Now, when $\lambda = 1$ the importance sampling weights all become $c_t = 1$ and we get

$$= \mathcal{R}(s_0, a_0) + \mathbb{E}_{a_t \sim \mu}\left[\sum_{t=1}^{H} \gamma^t(\mathcal{R}(s_t, a_t) + V^\pi(s_t) - Q(s_t, a_t)) + \gamma^{H+1}V^\pi(s_{H+1})\right]$$

$$= \mathcal{R}(s_0, a_0) + \mathbb{E}_{a_t \sim \mu}\left[\sum_{t=1}^{H} \gamma^t(\mathcal{R}(s_t, a_t) + \mathbb{E}_{\hat{a}_t \sim \pi}[Q(s_t, \hat{a}_t) - \alpha \log \pi(\hat{a}_t|s_t)] - Q(s_t, a_t))\right]$$
$$+ \gamma^{H+1}V^\pi(s_{H+1})$$

$$= \mathcal{R}(s_0, a_0) + \sum_{t=1}^{H} \gamma^t(\mathbb{E}_{a_t \sim \mu}[\mathcal{R}(s_t, a_t)] + \mathbb{E}_{\hat{a}_t \sim \pi}[Q(s_t, \hat{a}_t)]$$
$$- \mathbb{E}_{\hat{a}_t \sim \pi}[\alpha \log \pi(\hat{a}_t|s_t)] - \mathbb{E}_{a_t \sim \mu}[Q(s_t, a_t)]) + \gamma^{H+1}V^\pi(s_{H+1})$$

Since $\mu = \pi$, the expectations over $Q$ are identical and it further reduces to

$$= \mathcal{R}(s_0, a_0) + \sum_{t=1}^{H} \gamma^t(\mathbb{E}_{a_t \sim \pi}[\mathcal{R}(s_t, a_t)] - \mathbb{E}_{a_t \sim \pi}[\alpha \log \pi(a_t|s_t)]) + \gamma^{H+1}V^\pi(s_{H+1})$$

$$= \mathcal{R}(s_0, a_0) + \mathbb{E}_{a_t \sim \pi}\left[\sum_{t=1}^{H} \gamma^t(\mathcal{R}(s_t, a_t) - \alpha \log \pi(a_t|s_t)) + \gamma^{H+1}V(s_{H+1})\right].$$

Finally, we see that in the on-policy case with $\lambda = 1$, the Retrace target reduces to the regular SAC $H$-step value expansion target from Equation 3.

### A.3 EXPERIMENT DETAILS

This section provides detailed information regarding policy and $Q$-function networks, as well as the learned dynamics model. Further, we include hyperparameters of the training procedure.

**Policy Representation.** The policy is represented by a neural network. It consists of two hidden layers with 256 neurons each and ReLU activations. The network outputs a mean vector and a per-dimension standard deviation vector which are then used to parameterize a diagonal multivariate Gaussian.

**$Q$-function Representation.** To represent the $Q$-function we use a double $Q$ network. Each network has two hidden layers with 256 neurons each and ReLU activations and outputs a single scalar $Q$ value. The minimum of the two $Q$ networks is taken in critic and actor targets.

**Learned Dynamics Model.** The learned dynamics model is a reimplementation of the one introduced in Janner et al. (2019). It is an ensemble of 5 neural networks with 4 hidden layers with 256 neurons each and ReLU activations. To generate a prediction, one of the networks from the ensemble is sampled uniformly at random. The selected network then predicts mean and log variance over the change in state. This change is added to the current state to get the prediction for the next state.

**Hyperparameters for Training.** For a fair comparison, hyperparameters are shared across SAC and DDPG for the different expansion methods. In Table A.2 we report the hyperparameters across all experiments.

**Computation Times.** We report some of the computation times for the reader to get a better grasp of the computational complexity. Even though our GPU-only implementation using Brax (Freeman et al., 2021) is much more efficient than having a CPU-based simulator and transferring data back and forth to the GPU, the computational complexity of rolling out the physics simulator in the inner training loop, is still significant. This relates to complexity in computation time as well as VRAM requirements. Table A.3 shows the computation times on an NVIDIA A100 GPU.

Table A.2: Hyperparameters used for SAC and DDPG training. Note, that DDPG does not use the $\alpha$ learning rate due to its deterministic policy.

| Parameter | Inverted Pendulum | Cartpole | Hopper | Walker2d | HalfCheetah |
|---|---|---|---|---|---|
| policy learning rate | | | $3e^{-4}$ | | |
| critic learning rate | | | $3e^{-4}$ | | |
| $\alpha$ learning rate | | | $5e^{-5}$ | | |
| target network $\tau$ | | | 0.005 | | |
| Retrace $\lambda$ | | | 1 | | |
| number parallel envs | | 1 | | 128 | |
| min replay size | | 512 | | $2^{16}$ | |
| minibatch size | | | 256 | | 512 |
| discount $\gamma$ | 0.95 | | 0.99 | | 0.95 |
| action repeat | 4 | | 2 | | 1 |

Table A.3: Computation times of the training with oracle dynamics. The computational overhead dominates for short horizons already.

| H | 0 | 1 | 3 | 5 | 10 | 20 | 30 |
|---|---|---|---|---|---|---|---|
| **InvertedPendulum** | | | | | | | |
| SAC-CE | 6m | 8m | 10m | 12m | 18m | 28m | 42m |
| SAC-AE | 6m | 14m | 30m | 42m | 1h | 2h | 3h |
| Retrace | 6m | 6m | 7m | 8m | 10m | 14m | 18m |
| **Hopper & Walker2d** | | | | | | | |
| SAC-CE | 20m | 1h20m | 2h30m | 5h30m | 10h30m | 20h20m | 30h20m |
| SAC-AE | 20m | 9h 40m | 35h | 55h | 114h | 167h | 278h |
| Retrace | 20m | 20m | 22m | 23m | 27m | 30m | 40m |
| **HalfCheetah** | | | | | | | |
| SAC-CE | 40m | 3h15m | 5h 40m | 13h | 25h | 49h | 77h |
| SAC-AE | 40m | 30h | 62h | 100h | 168h | - | - |
| Retrace | 40m | 41m | 45m | 50m | 52m | 1h | 1h30m |

## A.4 TARGET DISTRIBUTIONS AND INCREASING VARIANCE

As mentioned in the main paper, SAC estimates $Q^H(s,a)$ by rolling out a model with a stochastic policy. This can be seen as traversing a Markov reward process for $H$ steps, for which it is well known that the target estimator's variance increases linearly with $H$. At the same time, the value expansion targets with oracle dynamics should model the true return distribution better. In order to confirm this, we compare the true return distribution, approximated by particles of infinite horizon Monte Carlo returns, to the targets produced by different rollout horizons using $D_W^H = \mathbb{E}_{s,a\sim\mathcal{D}}\big[D_W\big(\{Q_p^\infty(s,a)\}_{p=1\ldots P}\big|\{Q_p^H(s,a)\}_{p=1\ldots P}\big)\big]$, where $D_W$ is the Wasserstein distance (Villani, 2009), $P = 100$ particles per target (to model the distribution over targets). $Q^\infty(s,a)$ is the true Monte Carlo return to compare against (for computational reasons we use a rollout horizon of 300 steps). Finally, we evaluate the expectation of this estimator over $10^4$ samples $(s,a) \sim \mathcal{D}$ drawn from the replay buffer. We repeat this analysis for all horizons and 25 checkpoints along the vanilla SAC training. Figure A.1 2nd row shows that with increasing horizons $D_W^H$ decreases , i.e., as expected, longer rollouts do indeed capture the true return distribution better. The 3rd and 4th row of Figure A.1 depict the expectation of the target values particles over a replay buffer batch $\mathbb{E}_{s,a\sim\mathcal{D}}[\mathbb{E}[\{Q_p^H(s,a)\}_{p=1..P}]]$, and the corresponding variance $\mathbb{E}_{s,a\sim\mathcal{D}}[\mathrm{Var}[\{Q_p^H(s,a)\}_{p=1..P}]]$. The expected values of the target distribution $Q^H$ are very similar for different horizons. This observation means that the approximated $Q^0$-function already captures the true return very well and, therefore, a model-based expansion only marginally improves the targets in expectation. The 4th row shows, however, that the variance over targets increases by orders of magnitude with the increasing horizon (darker lines have larger variance).

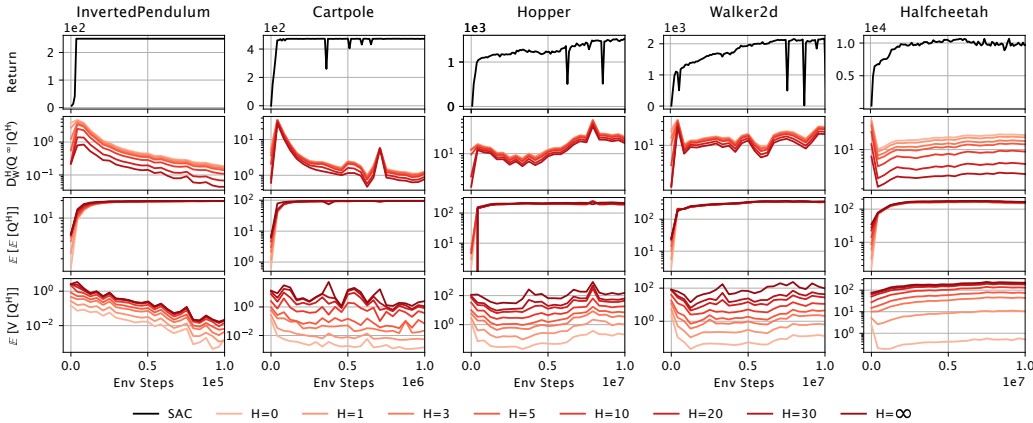

Figure A.1: Analysis of $H$-step targets for multiple checkpoints along the training process. The first row shows the undiscounted episodic return of SAC along training for one seed. The checkpoints along this single training run are used for the analyses in the following rows. The second row shows the Wasserstein distance between the sample-based target distribution predicted by the $H$-step estimator and the true target distribution represented by Monte Carlo samples. Rows three and four display the mean and variance over particles of the sample-based target distributions. The mean targets for different horizons are roughly equal, while the variance increases with the horizon.

## A.5 ANALYSIS OF DISCOUNT FACTOR

Another hypothesis for the source of the diminishing return could be the discount factor $\gamma$. Figure A.2 shows a line of initial experiments with an ablation over different discount factors on the InvertedPendulum environment. The phenomenon of diminishing returns is obvious in all experiments no matter the discount factor. We also do not see a correlation between the diminishing return and the discount factor $\gamma$ in these experiments. This leads us to believe that the discount factor is not the source of diminishing returns. Further experiments and a formal investigation are left to future work.

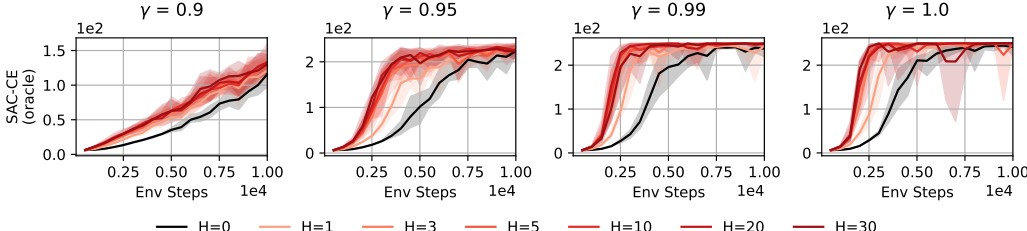

Figure A.2: Ablation of different discount factors $\gamma$ on the InvertedPendulum. The discount factor does not appear to influence the diminishing returns and, therefore, the discount factor does not seem to be the source of diminishing returns.

## A.6 GRADIENT EXPERIMENTS

Figures A.3, A.4, A.5 and A.6 show critic and actor gradients' mean and standard deviation for the different expansion methods of SAC and DDPG, respectively.

### A.6.1 SAC CRITIC EXPANSION GRADIENTS

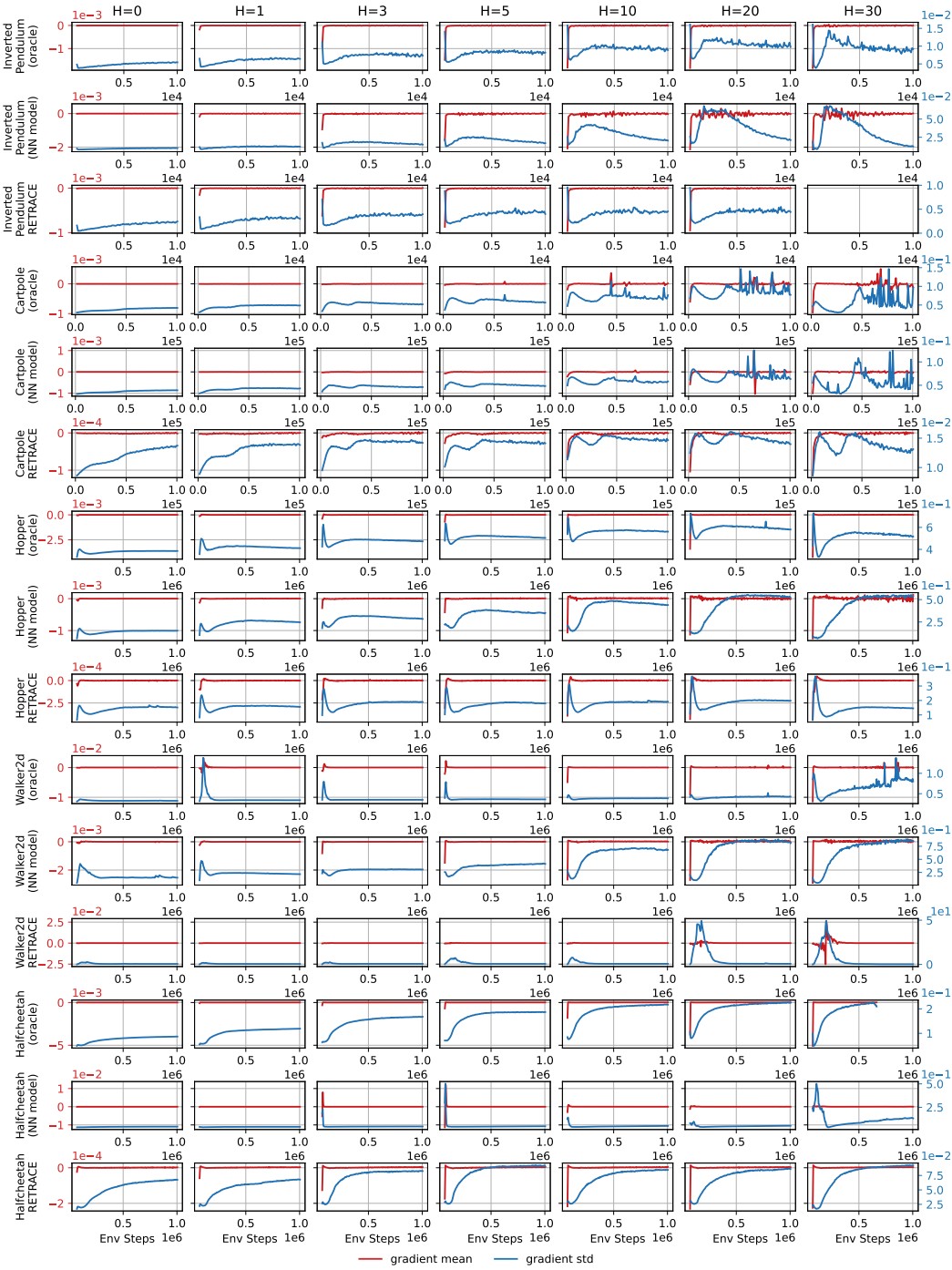

Figure A.3: These plots show the critic's gradients mean and standard deviation of SAC-CE for all environments and all rollout horizons. The values are in reasonable orders of magnitude and, therefore, the diminishing returns in SAC-CE cannot be explained via the critic gradients.

### A.6.2 SAC ACTOR EXPANSION GRADIENTS

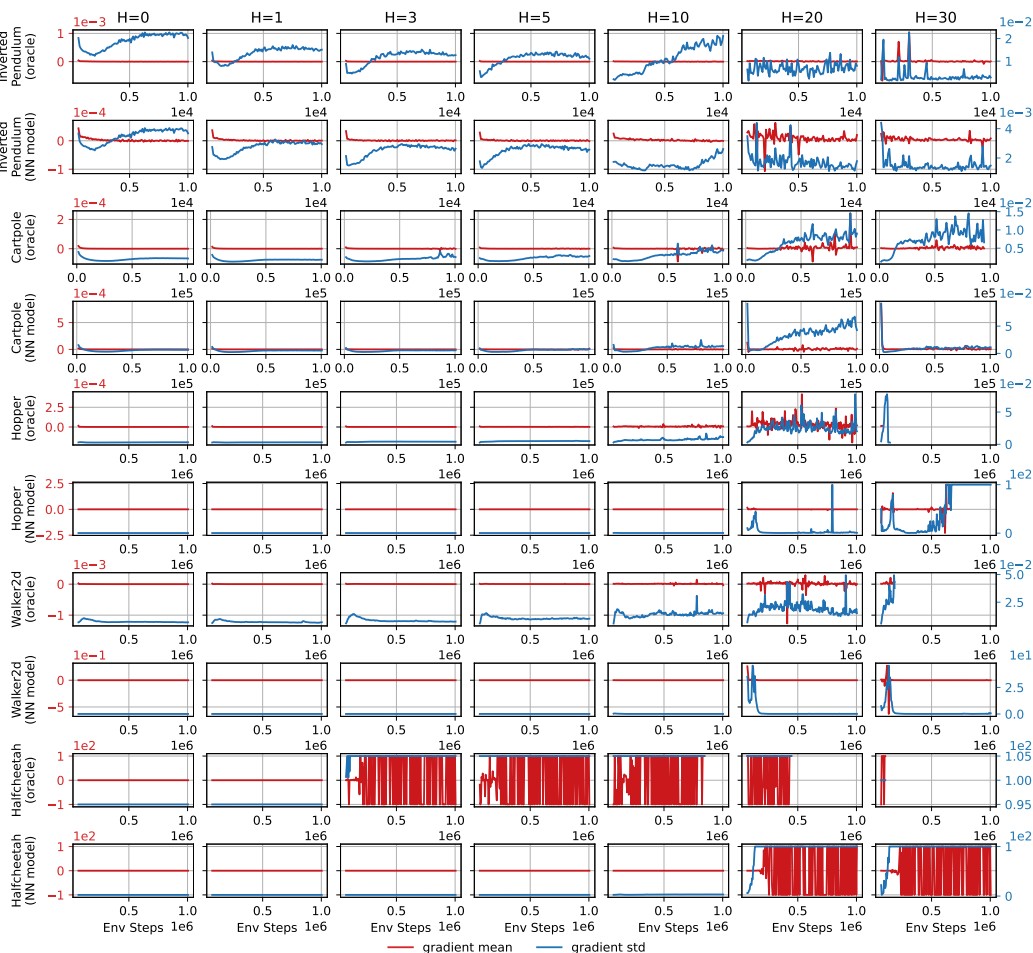

Figure A.4: These plots show the actor's gradients mean and standard deviation of SAC-AE for all environments and all rollout horizons. For readability, we clip large gradients to $10^2$ (Halfcheetah environment). Even though the gradients' mean have reasonable values (close to zero), except for the Halfcheetah environment, the standard deviation shows spikes and abnormal values for increasing rollout horizons, e.g., note the spikes in the standard deviation in the Hopper (NN model) experiments for horizons 20 and 30 (row 6, last two columns), which go up to the clipped value of $10^2$. These increase in variance can be correlated with the lower return in Figure 1 (row 5, col 3).

### A.6.3  DDPG CRITIC EXPANSION GRADIENTS

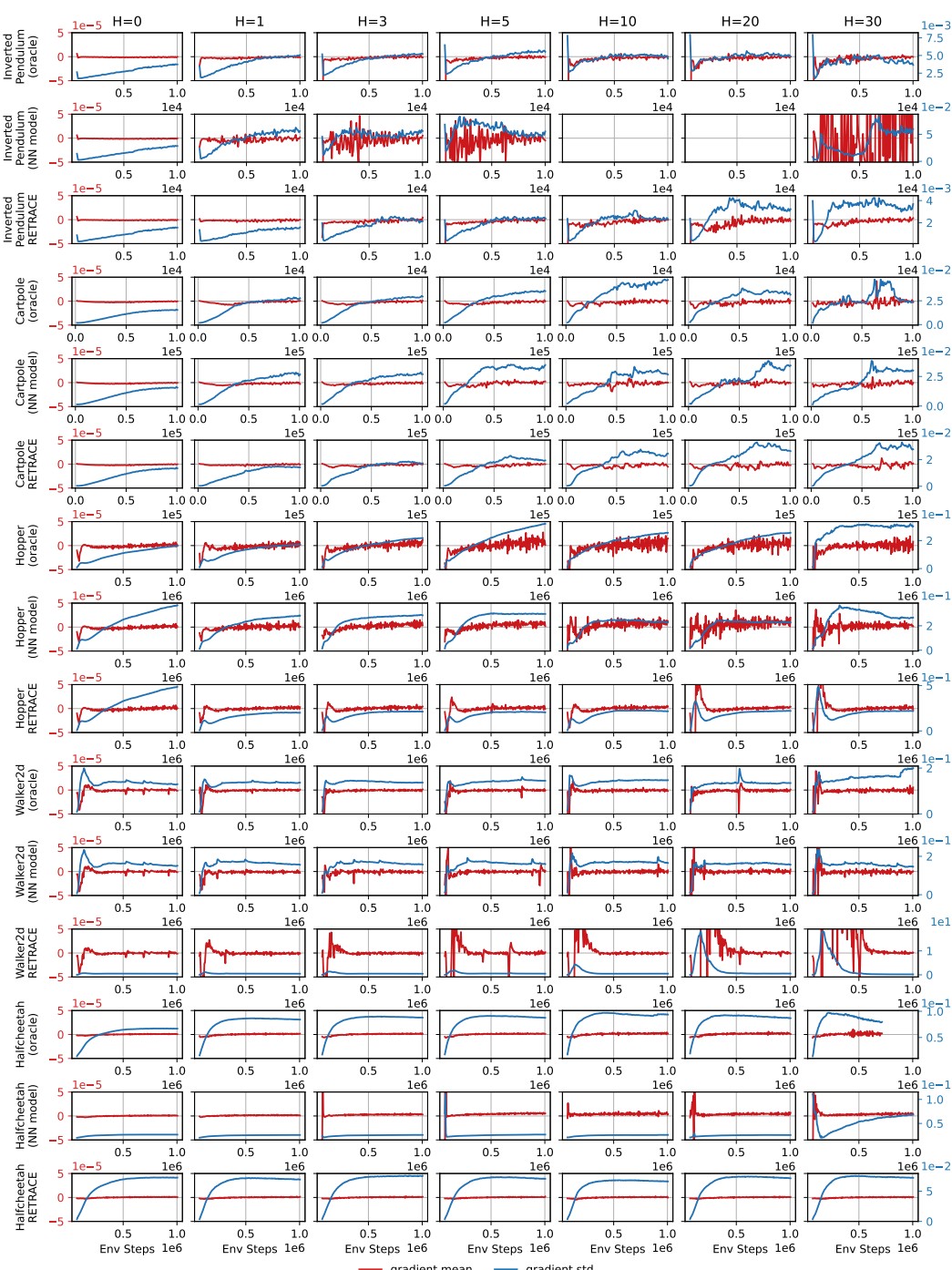

Figure A.5: These plots show the critic's gradients mean and standard deviation of DDPG-CE for all environments and all rollout horizons. The values are in reasonable orders of magnitude and, therefore, the diminishing returns in DDPG-CE cannot be explained via the critic gradients.

### A.6.4   DDPG ACTOR EXPANSION GRADIENTS

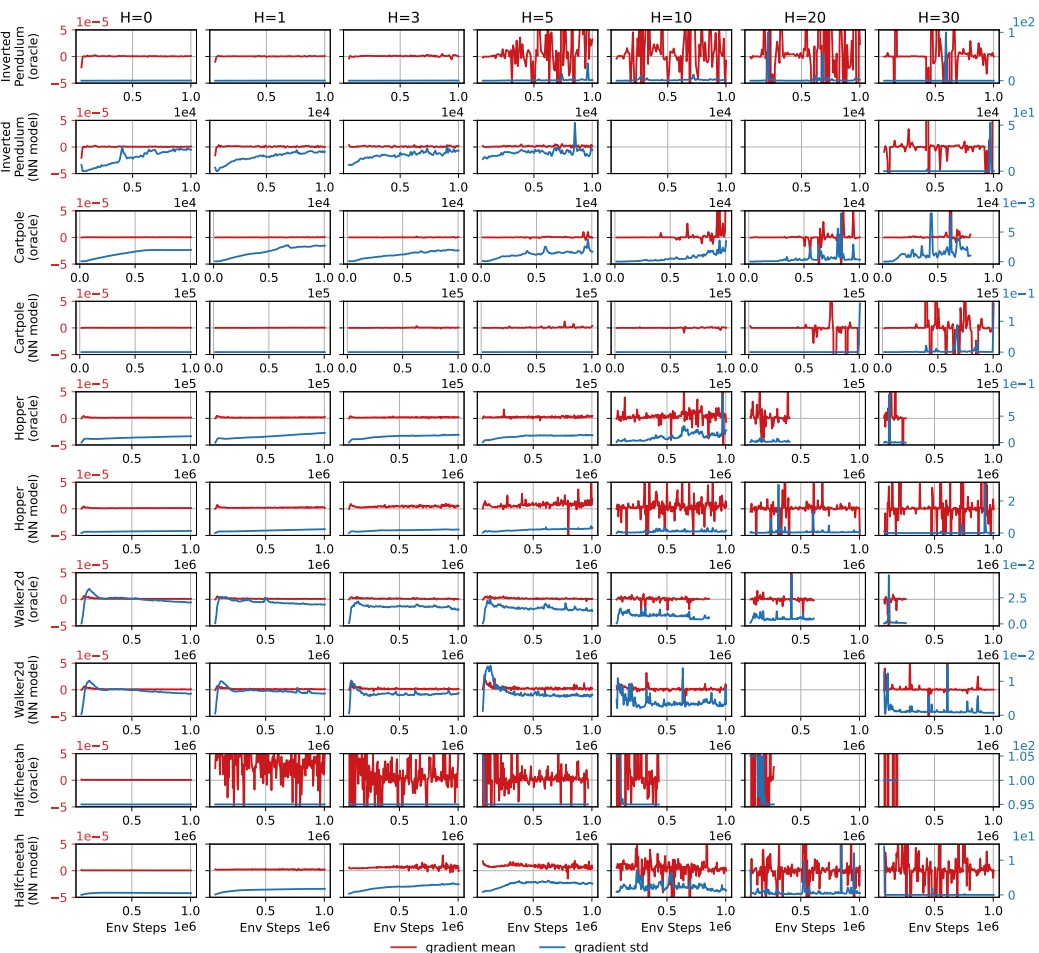

Figure A.6: These plots show the actor's gradients mean and standard deviation of DDPG-AE for all environments and all rollout horizons. For readability, we clip large gradients to $10^2$ (Halfcheetah environment).

