# OpenReview forum: "Diminishing Return of Value Expansion Methods in Model-Based Reinforcement Learning"
_ICLR.cc/2023/Conference — ICLR 2023 poster_

### Official Review · Reviewer_kpea · 2022-10-23

**Confidence:** 4
**Correctness:** 2
**Technical Novelty And Significance:** 3
**Empirical Novelty And Significance:** 3
**Recommendation:** 6

**Clarity, Quality, Novelty And Reproducibility:**

Writing:

- The paper is well-written in many places, but unfortunately one of the key parts, Section (2), is confusing.
- Notions are not very consistent: It defines $Q^{\pi, H}$ but never uses it. Eq (4) uses $V_t$ and $Q_t$ which are not defined.
- Please define the gradient mean and gradient variance in Section 3.3.

Typos:

- Page 4: "from real the environment" -> "from the real environment"

Reproducibility: The authors promise to release the code upon publication so I don't think it will be a concern.

Novelty: As far as I can see, this paper is novel. I'm not aware of existing work on analyzing how model accuracy affects sample efficiency. However, this paper might not be the first to reveal the phenomenon of diminishing returns, at least not for learned dynamics models.



**Strength And Weaknesses:**

Strength:

- The preliminary section covers many components and greatly helps the reader to understand the background. The paper is well organized.
- The problem this paper studies is indeed important. Diagonizing the bottleneck for model-based reinforcement learning algorithms can be quite difficult.

Weakness:

- There are a lot of missing details in Section 2, which makes it difficult to understand the exact algorithm this paper studies:
  - How is $V$-function in Eq (3) evaluated? Is it a different network than $Q$, as in [1], or simply sampling an action $a' \sim \pi(\cdot| s')$ and using $Q^\pi(s', a') - \alpha \log \pi(a' | s')$ as the result, as in [2]? More importantly, how is $V$-function evaluated in Eq (6)?
  - The Retrace target $Q^H_\text{retrace}$ doesn't encompass Equation (3): it doesn't take the entropy of the policy into consideration.
  - If $\lambda \neq 1$, $Q^H_\text{retrace} \neq Q^H$ and the paper doesn't provide the used $\lambda$.
- The paper only rules out the possibility of compounding error, but there are a lot of other factors which might affect the efficiency of model-based expansion. An important topic in estimating the value functions is how the bias-variance trade-off is taken into consideration. As I didn't see the paper makes effort to reduce the variance of $Q$ value estimation, I assume this paper only considers Monte-Carlo estimation, which is known to suffer from high variance. Retrace, which utilizes $V$-function to reduce variance and serves as a baseline in this paper, is another method to estimate the $Q$ values. One even simpler baseline is to average the $Q$ values over a few trajectories starting from the same $(s, a)$, given that the samples from the model are "free". Consider an extreme scenario, where we use the exact expectation of $Q^{\pi, H}(s, a)$ (so that the variance is reduced to 0) to do action-expansion or value-expansion. If the phenomenon of diminishing returns still exists in this scenario, I'll be convinced. The conclusion that I can draw from the paper is that with Monte-Carlo estimation of $Q$ values, an oracle dynamics model with a longer horizon might hurt.
- Even in the case we're using an oracle dynamics model, the diminishing return phenomenon is not surprising either. One can expect that the distribution of $Q^H(s, a)$ converges to $Q^\infty(s, a)$ at some rate when $H$ goes to infinity. (Indeed, this is what Figure 2 wants to show.) So even with a perfect $Q$ estimator with zero variance, the diminishing return may still exist. It will surprise me if the performance will drop, though.
- Maybe experiments show that 100 particles suffice to evaluate the Wasserstein distance between $Q^H(s, a)$ and $Q^\infty(s, a)$. Please also report the mean and variance of $Q^\infty(s, a)$ and $Q^H(s, a)$ so that the readers can get a sense of what the Wasserstein distance means.
- The author "conclude that the limitation of model-based value expansion methods is not so much the model accuracy of the learned models." However, in Figure 1.a,  SAC-CE (oracle model) performs significantly better than SAC-CE (NN model) in HalfCheetah, so the accuracy of the learned models **can** be a problem and can be an even greater problem than the choice of horizon under certain scenarios. What I can agree with is that it's not the only limitation.
- This paper doesn't have any theory and doesn't find the root causes for the phenomenon either.



[1]: Soft Actor-Critic: Off-Policy Maximum Entropy Deep Reinforcement Learning with a Stochastic Actor, Haarnoja. etc.
[2]: Soft Actor-Critic Algorithms and Applications, Haarnoja. etc.



**Summary Of The Paper:**

This paper studies the problem of how model accuracy relates to sample efficiency a class of model-based reinforcement learning algorithms, that is, model-based value expansion algorithms. The paper conducts experiments to show that the improvement of using the oracle dynamics model over a learned dynamics model is not too much, and there is a phenomenon of diminishing return for both, which means short horizons suffice to achieve the best sample efficiency, while model-free baselines are only slightly worse. Thus, the authors conclude that model-based value expansion algorithms are not limited by the accuracy of the learned dynamics model.

**Summary Of The Review:**

My major concerns with this paper are that the variance of the $Q$ value estimation is not taken into consideration. The writing can be more precise and less confusing. I don't recommend acceptance in the current form but am glad to increase my score if my concerns can be addressed.

---
Update: As the authors provide more experiments on the bias-variance trade-off, addressing my major concern, I'd like to increase the score to 6.

---

> ### Author Response · Authors · 2022-11-17
> **Response to Reviewer kpea**
>
> Thank you for your positive comments and the acknowledgement of the importance of the problem that this paper studies. Thank you also for the constructive feedback and the good questions. We have uploaded an updated version of the paper where we attempt to address all your concerns.
>
> ### Variance Reduction
> One reason for including the DDPG experiments is to show the diminishing return even in the absence of target variance. As DDPG has a deterministic policy, and the oracle dynamics are deterministic too, the Monte Carlo rollouts do not have any variance.
>
> Based on your suggestion for active variance reduction, we have also conducted and included further experiments for SAC-CE and SAC-AE. For these experiments, we average over multiple particles per target to reduce variance in the targets. The results are qualitatively identical to the initial experiments with the single sample Monte Carlo estimates.
>
> We have added an additional section where we discuss those results. Based on the particle experiments and DDPG we conclude, that the increased variance over longer horizons in SAC-CE / SAC-AE is not the cause of the diminishing returns.
>
> ### Missing Details in Section 2
> We only learn a double Q function and not a separate value function. The value function is evaluated by sampling an action $a\sim\pi$ and using $Q(s,a) - \alpha\log\pi(a|s)$ as in [1]. We have updated the paper to state this clearly.
>
> We have reviewed and cleaned up the notation and are sorry about the confusion it caused.
>
> ### RETRACE for mu=pi
> We use \lambda=1 in the entire paper and have added that detail in the text.
> We further added a detailed derivation demonstrating how Eq. 6 reduces to the $Q^H$ target (Eq. 3) in the on-policy case in the Appendix. In that case, the entropy penalty comes out of the term $V(s,a) - Q(s,a) = - \alpha\log\pi(a|s)$.
>
> ### Wasserstein plot
> We have included Q^\inf(s,a) as well as the mean of the particle-based distributions.
>
> [1] Soft Actor-Critic Algorithms and Applications, Haarnoja. et. al
>
> We want to thank you again for your good review and the constructive feedback and questions.

---

### Official Review · Reviewer_5jBe · 2022-10-24

**Confidence:** 4
**Correctness:** 4
**Technical Novelty And Significance:** 3
**Empirical Novelty And Significance:** 4
**Recommendation:** 8

**Clarity, Quality, Novelty And Reproducibility:**

- Quality: medium-high: the quality is overall good, but the weakness above regarding existing implementations of algorithms raises the question of "how can we be certain that there is no bug in the experiments?"
- Clarity: high. The paper does an amazing job at explaining model-based RL and its variants, and where rollouts can be used. This is material to be used in a classroom. The rest of the paper flows well and the results are presented in a clear way. Small remark: the figures would benefit from one extra sentence that states what we have to see in them. Instead of "these are the variances of the losses", it is useful to have "this are the variances of the losses. They seem normal and longer horizons do not appear to have an effect of them."
- Originality: high. This work questions an aspect of model-based RL that was not questioned before, and found a problem.
- Reproducibility: medium-high. It would be important to have the source-code used in the paper to inspect it for correctness.

**Strength And Weaknesses:**

Strengths:

- The paper contains information that clearly needs to be known by the RL community. Recent algorithms, such as Dreamer and MuZero, leverage models and do planning/rollouts in them. It is important to know that there is possibly a problem in RL that prevents these models, as good as they may be, from benefiting the agent.

Small weakness:

- Some directions have been pursued as to why longer horizons are not always beneficial, but these directions did not bear fruit. The paper therefore identifies a problem without a direction for solving it.
- The experiments are based on SAC and DDPG, with their losses modified to include model-based rollouts (in ways explained in the paper). It would have been very interesting to also see experiments with existing implementations of SOTA algorithms, such as DreamerV2, and with plugging the oracle model in them. This would have allowed to be certain that the diminishing return of H does not come from an implementation error in this paper.

**Summary Of The Paper:**

This paper analyses model-based RL methods and studies the impact of the model-based rollout horizon H. The main finding of the paper is that longer horizon do not guarantee better results (they have diminishing returns, and may even be detrimental). This is even true for oracle models (perfect models!), so it is not caused by compounding model errors.

Half the experiments focus on identifying the above effect (larger H is not always better), and the other half of the experiments try to find why longer horizons do not always perform well. No answer is found for that last question, and the paper concludes that future research needs to focus on why longer horizons do not benefit model-based RL.

**Summary Of The Review:**

Clear information that needs to be available to the RL community. There is only a minor problem of whether model-based RL is limited in some way, or there is a bug somewhere.

---

> ### Author Response · Authors · 2022-11-17
> **Response to Reviewer 5jBe**
>
> Thank you for your positive feedback on our paper and the acknowledgement of the importance of the research question and results.
>
> We agree that it would be interesting to also investigate SOTA algorithms like the mentioned DreamerV2 in future work.

---

### Official Review · Reviewer_vr7K · 2022-10-25

**Confidence:** 2
**Correctness:** 3
**Technical Novelty And Significance:** 2
**Empirical Novelty And Significance:** 2
**Recommendation:** 6

**Clarity, Quality, Novelty And Reproducibility:**

This paper is well-written and easy to follow. The experimental results are extensive and seem to be solid.

**Strength And Weaknesses:**

The results from the paper are extensive and are relevant to the model-based RL community. And the observations that a learned model achieves efficiency similar to a perfect model and RETRACE is a strong baseline for model-based algorithms are interesting.

However, I think the empirical results are kind of expected. The authors focused on Q function estimation using a single trajectory trajectory (Eq (3)), which is known to suffer from large variance when horizon increases. Many existing methods are designed to specifically tackle this problem [1][2]. In the experiments, the authors can explore this direction by reducing the variance in the true and learned dynamics model and using these variance reduction methods under the value expansion framework. Furthermore, the phenomenon of diminishing returns can be simply attributed to the exponential decay due to the reward discount factor.


Would the authors please clarify the sentence on In page 7, “we show that the critic gradients are probably not the source of the diminishing returns.”?

Typo: second line in Introduction, samples one -> sample on

[1] Grathwohl, Will, et al. "Backpropagation through the void: Optimizing control variates for black-box gradient estimation." arXiv preprint arXiv:1711.00123 (2017).

[2] Cheng, Ching-An, Xinyan Yan, and Byron Boots. "Trajectory-wise control variates for variance reduction in policy gradient methods." Conference on Robot Learning. PMLR, 2020.


**Summary Of The Paper:**

The authors conduct an extensive empirical study using simulated control tasks on a family of model-based RL algorithms to answer the following question: how much more sample efficiency can be gained by improving the learned dynamics models. The results from the experiments are contrary to the common belief that regards the accuracy of the dynamics model as a limitation of model-based approaches. The experimental results show that learned models provide on-par improvements compared to perfect models and the improvements diminishes with increasing horizon. Furthermore, the off-policy method RETRACE that does not rely on dynamics models and thus has better computational complexity archives similar sample efficiency.


**Summary Of The Review:**

Due to the slight lack in significance as discussed above,

---

> ### Author Response · Authors · 2022-11-17
> **Response to Reviewer vr7K**
>
> Thank you for your review and for acknowledging the importance of the paper and results to the community. We uploaded an updated version of the paper to address your questions and concerns.
>
> In this paper, we wanted to stress that a lot of current research focuses on building more accurate dynamics models to be used in value expansion, and our experiments show that it’s not clear if the added computational effort brings a benefit in comparison to model-free methods like RETRACE.
> However, it is the case that a lot of current research focuses on building more accurate dynamics models. While we show (1) that current models can oftentimes not be significantly outperformed by even oracle dynamics. (2) that there appears to be an upper limit for value expansion methods which is rooted in the effect of diminishing return.
>
> ### Target Variance
> To address the target variance, we have expanded the discussion about DDPG (which does not have any variance in its targets). And we have further added an ablation for SAC with multiple particles per target to reduce the variance. In both cases, we still see the same effect.
>
> The references you referred to are related to reducing the variance of estimators to compute the gradient of an expectation wrt the distributional parameters, for e.g. wrt to the mean and variance of a stochastic Gaussian policy. However, we do not see how they apply to estimate the critic target, as we are not estimating the gradient, but rather a single scalar.
>
> ### Discount Factor as Cause of Diminishing Returns
> We have further added an ablation with multiple discount factors in the Appendix. In those experiments, we could find, that the discount factor had any influence on the diminishing returns. Based on this, we draw the conclusion, that the discount factor is not the cause of diminishing returns.
>
> ### Would the authors please clarify the sentence on In page 7: There is no clear signal in the gradients which would indicate that they would be the root cause of the diminishing returns.
> The critic’s gradient means and standard deviations are in a reasonable range for all environments and algorithms. The mean is close to zero, and the standard deviation is between 10^-3 and 10^-1. Therefore, for the critic expansion, we believe the gradients are not the issue for diminishing returns.
>
> We want to thank you again for your review.

---

> > ### Comment · Reviewer_vr7K · 2022-12-02
> > **Thank authors for the response**
> >
> > I really appreciate the authors' response and revision to the paper. They answered my questions.
> >
> > There are definitely interesting take-aways from paper. The fact that a class of common model-based methods can not benefit much from true transition model is surprising. This paper can bring this issue to researchers' attention and stimulate the study on new model-based methods. Therefore, I increased my score.
> >
> > However, on the other hand, the diminishing return phenomenon for methods that use model for improving value function estimation is also not surprising. The value in true dynamics is limited by off-policy samples and function approximation. I encourage the authors to come up with other hypotheses and verify them theoretically and empirically, potentially beyond the continuous state and action control domain.

---

> > > ### Author Response · Authors · 2022-12-05
> > > **Thank reviewer vr7k**
> > >
> > > We thank the reviewer for their valuable feedback and for increasing the score of the paper.

---

### Official Review · Reviewer_Q2WM · 2022-10-31

**Confidence:** 4
**Correctness:** 3
**Technical Novelty And Significance:** 3
**Empirical Novelty And Significance:** 3
**Recommendation:** 6

**Clarity, Quality, Novelty And Reproducibility:**

I appreciated the paper's aim to thoroughly investigate a single, well defined question. I generally found the writing to be clear. I think the question studied here is highly relevant to the RL community, and I think careful empirical studies such as these into such questions are potentially highly valuable. However, I believe there currently some issues with reproducibility, which I hope the authors can address.

The authors commit to open-sourcing the code. However, there is very little discussion of hyperparameter settings and other experimental details in the paper. In particular, no hyperparameter settings are described for the SAC and DDPG agents, or for the RETRACE baseline (e.g. lambda), or for the neural network dynamics model, as far as I am aware. These details will be crucial for reproducibility of the paper, and just as importantly, the interpretation of the results presented in the paper.

As an example, it is difficult to compare the RETRACE baseline with the model-based approaches without knowing what kinds of hyperparameter sweeps were carried out in the course of the research.

In addition, there don't appear to be any details of the neural network dynamics model used in the paper, but details on this (is it a deterministic or stochastic model) are crucial for understanding the results.

Other important experimental details "in practice, we use a large rollout horizon based on the discount factor γ" are also currently omitted, and should be included in the appendix.

**Strength And Weaknesses:**

This is an empirical paper, which seeks to address a common belief in model-based RL: that model accuracy is a bottleneck to performance in value-expansion methods in RL, prohibiting us from doing longer-horizon expansions. The singularity of the paper's focus is a strength, and I found the work to be clearly presented. Papers such as this have the potential to be highly useful to the community, in carefully examining commonly-held beliefs, and impacting the practice of model-based RL in the future. However, I do have several comments on areas where I believe the paper currently falls short, which is the basis for my current rating. I would be open to revising this rating based on the authors' response.

_Scope_

I appreciated the inclusion of several different agents, and several environments. While it is always possible to include more comparisons in a paper such as this, and a line needs to be drawn somewhere, I think there are a few important comparisons which, even if not included in the paper, should be discussed.

Deterministic policies. Although the paper considers the DDPG agent in Figure 1, this agent does not seem to be investigated further in the paper. However, I think analyzing DDPG gradients in a similar way to the SAC analysis would be an interesting additional data point for the paper. If the DDPG policy and model are deterministic, this removes all stochasticity from the gradients, and therefore potentially provides a more nuanced answer to the question: "do diminishing returns stem from increased gradient variance", which is not currently answered conclusively in the paper.

Error in bootstrapped value function. As far as I can tell, all bootstrapped value functions used are those learnt by the SAC agents. Since one of the main hypotheses as to what drives the change in performance as a function H is a bias/variance trade-off for gradient estimation, it would have been interesting to see the effects of more/less accurate bootstrap value estimates on performance.

Other types of environment. The paper focuses on deterministic, continuous control tasks. I would not necessarily expect experiments beyond this in a conference paper, but some discussion as to whether the authors expect the findings to generalise beyond this would be very valuable. For example, could stochastic transitions affect the bias/variance trade-off and lead to very different performance as a function of H.

Further values of H. The largest value of H considered in the paper is 30. With a discount factor of 0.99, this still attaches a weight of 0.99^30 = 0.74 to the bootstrapped value function. It would have been interesting to see results with value expansion closer to Monte Carlo simulation (i.e., with a value of H so that gamma^H is reasonably close to 0).

To emphasize, I don't think it's necessary to run further experiments relating to each of these comparisons (although doing so would obviously increase the impact of the paper, and the weight of any conclusions drawn), but I think discussion, and possible qualification, of the paper's findings with these comparisons in mind would strengthen the paper, and make it more useful for the community in general.



Some additional comments for each section are provided below

_Section 2_

In the definition of Markov decision processes given here, some non-canonical choices are made: state/action spaces as subsets of R^n, transitions having probability density functions (which is actually not satisfied by the (deterministic) continuous control tasks considered later in the paper). How important are these assumptions?

Eqn (2): Should there be a stop gradient around V^pi(s') in this case?

Just below Eqn (6), the authors claim that when mu = pi, the RETRACE estimate reduces to the H-step action-value target. This is not true, because c_{t-1}V_t - c_t Q_t, which appears in Eqn (6), does not evaluate to 0.

_Section 3.2_

I am unsure exactly what is intended to be taken away from the Wasserstein comparison. It is clear that the distance should go to 0 as H increases, but it is not clear to me why these Wasserstein distances are of direct importance to studying the source of diminishing returns.

The comparison of estimator variance as H varies feels more related to the topic of the paper, but further discussion would be valuable here: while variance increases with H, is the increase in variance correlated with the extent of diminishing returns observed?

_Section 3.3_

I found the results of this section very difficult to interpret. As far as I can see, the authors do not define the mean and standard deviation of the gradients here (is this on a per-coordinate basis?) Additionally, the scale for the gradient means the y-axis seems too large, so that the curves essentially look like the line y=0 in most panels.

The authors reach the conclusion that gradient variance may contribute to the poor performance of large horizons in actor-expansion methods, but is not likely to be the cause of diminishing returns in critic-expansion methods. More discussion of these results would strengthen the paper (what makes the variance so catastrophic in some environments like HalfCheetah, and not in others)? If gradient variance doesn't explain diminishing returns for critic-expansion methods, what alternative hypotheses should be investigated further?

_Model-free vs. model-based comparison_

One of the main takeaways that the authors present is that the model-free expansion based on RETRACE is competitive to model-based methods. I agree with this conclusion when the number of learner steps is taken to be equal in both cases, but presumably one of the main advantages of using model-based expansion is that if the computational budget is available, more updates can be performed by sampling new experience from the model (while RETRACE is constrained just to use collected trajectories). This aspect of the comparison seems to be absent from the paper currently, and I think should at least be mentioned as another possible reason to prefer model-based value expansion over model-free methods.

_Other comments_

p1: "samples one" -> "samples from"
There are a few instances of citations to recent papers for well established concepts e.g. Deisenroth et al. (2013) for model-based RL, and (Peyré & Cuturi) (2019) for the Wasserstein distance.


**Summary Of The Paper:**

This paper has a clear focus, aiming to address the question "is model accuracy a bottleneck to value-expansion methods in RL?" The paper approaches this by studying the performance of two algorithm families, SAC and DDPG, in several continuous control tasks, using both a learnt dynamics model, and an oracle (exact) dynamics model.



--- POST-REBUTTAL

I thank the authors for their detailed response.

My initial review mentioned that this is a potentially impactful contribution that aims to address an important question in model-based reinforcement learning. I highlighted several reservations with the initial draft, including: (i) lack of architecture descriptions/hyperparameters/general concerns around reproducibility, (ii) the potential for further experiments to strengthen the evidence for the main hypothesis claimed in the paper, and (iii) the specificity of the findings to the agents/environments investigated in the paper.

In their response, the authors have included more qualifications in the paper regarding (iii).

The inclusion of the further DDPG experiments is a good step towards checking the bias-variance explanation of their findings with regard to (ii) as well; this definitely strengthens the paper in my view. I think there is still scope for further experimentation here, but this shouldn't be a barrier to acceptance in my view. In particular, the authors mentioned that evaluating the bias-variance trade-off further, or the impact of an inaccurate value function, by extending the roll-out horizon beyond H=30, was computationally intractable. The inclusion of the run-time table in the appendix makes this clear, and I think adds useful context to the experiments carried out in the main paper. However, using a smaller discount factor would make H=30 sufficient to essentially eliminate the bias from the inaccurate value function, and this could have been an interesting additional experiment to run.

Regarding (i), the authors have included more details regarding agent hyperparameters, and the neural-network-based dynamics model. I think this is now at the level of detail where reproducibility is realistic. The paper could still be strengthened by including more details about any hyperparameter sweeps carried out e.g. in training the NN dynamics model.

Aside from these larger points, the authors' response also clarified many of the more minor queries I initially raised.

Based on the factors above, I have increased my rating for the paper, and would argue for it be accepted.

I think there is still scope for the clarity of the paper to further improved. One aspect in particular that came up in discussions with other reviewers and the meta-reviewer is that the messaging of the core take-aways of the paper are perhaps a bit unclear. In some sense, it is arguably not surprising that there are diminishing returns from increasing the rollout horizon (once H >> 1/(1 - gamma), for example, the bias of the value function has essentially been eliminated). In contrast, the observation that moving from a learnt model to an oracle model has limited impact on algorithm performance seems like a much more impactful take-away.

**Summary Of The Review:**

In my view, this is an interesting paper that tackles an important question. I think it has the potential to be impactful in the RL community, but I currently see several issues that prevent me from recommending acceptance, including (i) the lack of clarity in the way the results are communicated in Section 3.3, which are crucial to the main findings of the paper, (ii) a discussion of the generality of the findings, and (iii) full experimental details. I would be open to potentially adjusting my rating in response to the authors' rebuttal.

---

> ### Author Response · Authors · 2022-11-17
> **Response to Reviewer Q2WM**
>
> We thank the reviewer for their positive comments on our work and for recognizing the importance of the question and our results. We have updated the paper and include additional experiments to address your questions and concerns. We have widened the scope of the paper both in terms of discussion and additional experiments as outlined below:
>
> ### Deterministic Policies
> We extend the discussion about DDPG. Further, we have added the DDPG gradients to the Appendix. Due to the combination of the deterministic policy and the dynamics, there is no variance in the DDPG targets which shows, that the diminishing return does not stem from target variance.
>
> ### Variance Reduction in the Targets
> We include additional SAC experiments where we reduce the variance by averaging over multiple particles for each target. The resulting training curves from this experiment are qualitatively identical to the ones without variance reduction. We conclude, that the increasing target variance for longer horizons does not impact the diminishing returns.
>
> ### Generality of the Findings
> We have adjusted the paper to include a discussion regarding the generality of the findings.
>
> We show that model-based value expansion does not strictly improve the sample efficiency but appears to have an upper limit which is quickly approached with short horizons already. In the case of actor expansion, longer horizons can even decrease performance due to gradient problems.
> The results we present focus on continuous state-action control tasks with deterministic dynamics. Therefore, we cannot claim generality of those results towards other problem settings. The investigation of the transferability of the results towards other types of environments and tasks is left for future work.
>
> ### Larger values of H
> We agree that much larger rollout horizons would be interesting to analyze as well. However, the computational complexity of unrolling the oracle dynamics model in the inner training loop is significant. We have included a Table of computation times in the Appendix. These experiments would need to be performed on different tasks / with different simulators.
>
> ### Definitions and Notation in Section 2.
> We removed both assumptions as they were not important. Furthermore, we cleaned up and clarified the notation.
>
> ### RETRACE for mu=pi
> We added a detailed derivation demonstrating how Eq. 6 reduces to the Q^H target (Eq. 3) in the on-policy case into the Appendix.
>
> ### Intend behind Wasserstein Analysis
> One common assumption is that a better approximation of the true return distribution will lead to faster learning, i.e. more accurate targets result in faster learning. The reason for including the Wasserstein plots is to show that the targets produced by longer horizons indeed do better capture the true return distribution. Given that the H-step targets are bootstrapped with a learned terminal Q-function, it could have been possible that that was not the case. There could have been a signal in those plots with regard to the diminishing returns, which is why we include them.
>
> ### Correlating Increasing Variance with Diminishing Returns
> Correlating the increase in variance with diminishing returns is an interesting question. We have attempted to find a correlation between the increase in variance and the extent of diminishing returns. In the short rebuttal phase, our preliminary results did not show a correlation. But we will continue to explore this question further. For now, we have addressed this in the discussion.
>
> ### Gradient Mean and Variance
> We define the mean and variance over all parameter gradients of the actor/critic and improved the writing in the paper to make this more clear.
> We have also adjusted the scaling of the plots and extended the discussion.
> And we have extended the discussion with regard to the gradients.
>
> ### Model-free vs. model-based comparison
> We have included a further discussion with regards to the advantages of model-based RL over model-free more, to have a fair comparison.
>
> We want to thank you again for your good review and the amount of constructive feedback and questions.

---

> > ### Comment · Reviewer_Q2WM · 2022-11-23
> > **Response to authors**
> >
> > I thank the authors for their response. In my view, the edits they have made to the paper have broadly improved the clarity of the paper, and I think the extra experiments, including the DDPG deterministic policy experiments, strengthen the conclusions reached. Several additional comments/questions in response are included below.
> >
> > RETRACE equivalence. The argument given in Appendix A.2 is showing that the operator corresponding to the RETRACE update (i.e. with an expectation over the sampled trajectory) matches the corresponding H-step return operator. This matches the argument given in the original paper by Munos et al. (2016), albeit without the entropy regularization terms.
> >
> > However, if I understand correctly, the authors are making a different claim in the main paper, below Eqn (6): not that the operators for RETRACE and H-step returns are equivalent, but that the trajectory-based approximations (without an expectation over the sampled trajectory) are equivalent. This is not true, as mentioned in the original review.
> >
> > Experiment set-up. The inclusion of some of the training details in Appendix A.3 improves the paper, although there is still scope to provide further details here that are important for interpreting the experimental results, such as how the quoted hyperparameters/network architectures were selected?

---

> > > ### Author Response · Authors · 2022-11-24
> > > **Response to Reviewer Q2WM**
> > >
> > > >I thank the authors for their response. In my view, the edits they have made to the paper have broadly improved the clarity of the paper, and I think the extra experiments, including the DDPG deterministic policy experiments, strengthen the conclusions reached. Several additional comments/questions in response are included below.
> > >
> > > We thank the reviewer for the positive feedback on our changes to the paper.
> > >
> > > >RETRACE equivalence. The argument given in Appendix A.2 is showing that the operator corresponding to the RETRACE update (i.e. with an expectation over the sampled trajectory) matches the corresponding H-step return operator. This matches the argument given in the original paper by Munos et al. (2016), albeit without the entropy regularization terms.
> > > >
> > > >However, if I understand correctly, the authors are making a different claim in the main paper, below Eqn (6): not that the operators for RETRACE and H-step returns are equivalent, but that the trajectory-based approximations (without an expectation over the sampled trajectory) are equivalent. This is not true, as mentioned in the original review.
> > >
> > > It is true that the RETRACE operator and the H-step return are equivalent in expectation, as shown in Appendix A.2. However, their single-sample trajectory-based approximations are indeed not necessarily identical. We will state it more clearly in the final version, and explain that our overloaded notation is rather for convenience and should not be misinterpreted. We will furthermore clarify that our argument matches the one from Munos et al. (2016) but with the added entropy regularization term.
> > >
> > > We would like to point out that this relationship between RETRACE and the H-step return is highlighted to form an intuitive understanding, because we feel that powerful off-policy methods like RETRACE get little attention in the related model-based value expansion literature. This discussion, however, does not impact the experimental results of the paper itself.
> > >
> > > We are happy to adapt this section of the paper to be more precise about the relationship, to more clearly state that it only holds in expectation and is presented to form an intuition.
> > >
> > > >Experiment set-up. The inclusion of some of the training details in Appendix A.3 improves the paper, although there is still scope to provide further details here that are important for interpreting the experimental results, such as how the quoted hyperparameters/network architectures were selected?
> > >
> > > We are happy to add further clarifications regarding this to the Appendix. The implementation choices and hyperparameters are based on the ones commonly found in the related literature (e.g. https://github.com/jannerm/mbpo, https://github.com/google/brax,).
> > > This was done, because on the one hand, in related work, these hyperparameter setting have proven to be well suited for the tasks. And, on the other hand, due the computational complexity of all of the experiments (Table A.3) our computational budget would not allow to perform large hyperparameter sweeps for all value expansion methods and rollout horizons.

---

### Author Response · Authors · 2022-11-17
**General Response**

We thank all the reviewers for their positive and detailed feedback and their good questions. We are pleased that the reviewers acknowledged the relevance of our paper’s research question and results.

- **q2wm**
  - The question studied here is highly relevant to the RL community.
  - Careful empirical studies such as these into such questions are potentially highly valuable.

- **vr7k**
  - The results from the paper are extensive and are relevant to the model-based RL community.

- **5jbe**
  - The paper contains information that clearly needs to be known by the RL community.
  - This work questions an aspect of model-based RL that was not questioned before and found a problem.

- **kpea**
  - The problem this paper studies is indeed important. Diagonizing the bottleneck for model-based reinforcement learning algorithms can be quite difficult.

We are especially pleased about positive feedback regarding the clarity, focus and writing of the paper:
- **5jbe**: The paper does an amazing job at explaining model-based RL and its variants, and where rollouts can be used. This is material to be used in a classroom.
- **vr7k**: This paper is well-written and easy to follow. The experimental results are extensive and seem to be solid.
- **q2wm**: The singularity of the paper's focus is a strength, and I found the work to be clearly presented.
- **kpea**: The preliminary section covers many components and greatly helps the reader to understand the background.

The reviewers also raise several good questions. There were three major points the reviews agreed upon, which we want to answer here: Bias-Variance tradeoff, missing experiment information for reproducibility and additional discussion.

### 1. Bias-Variance Tradeoff
To evaluate the bias-variance trade-off, we have performed two experiments.
- We have used DDPG as an additional algorithm to SAC as for DDPG there is no variance, due to the deterministic policy and deterministic simulator. The diminishing returns are also present for DDPG.
- We include additional SAC experiments that average the targets over 30 particles to reduce the variance. Also in these experiments, the diminishing returns are observable not significantly different to the experiments without averaging. Note, that due to the short rebuttal phase and the computational complexity, not alle horizons and environments did finish yet. All experiments will be included in the final paper.

As the Bias-Variance Tradeoff was not extensively discussed in our original paper, we have reworked the paper to explicitly discuss the bias-variance tradeoff.

### 2. Experimental Details
To improve the reproducibility of the paper we have
- added clarifications to the main paper,
- added a section on the experimental details to the appendix and
- submitted the code here in the supplementary material.

### 3. Additional Discussion
We have added additional discussion and details regarding
- the generality of the findings
- Bias-Variance tradeoff
- DDPG gradients

We want to again thank all reviewers for their time and effort.

---

### Decision · Program_Chairs · 2023-01-20

**Decision:**

Accept: poster

**Justification For Why Not Higher Score:**

The paper studies a somewhat narrow phenomenon in model-based RL, and may not be of interest to everyone in the full ICLR community.

**Justification For Why Not Lower Score:**

The paper studies an important and counter-intuitive phenomenon in model-based RL which is important for the MBRL community to know about.

**Metareview: Summary, Strengths And Weaknesses:**

This paper asks the question: does compounding model-error lead to diminishing returns data efficiency in model-based RL? The paper focuses in particular on a specific class of MBRL algorithms, model-based value expansion, and looks at two ways of incorporating the model-based estimate into learning (either for the actor update or the critic update) in two different learning algorithms (SAC and DDPG). In both cases, using an oracle model does not substantially improve performance over a learned model, and longer rollouts using the oracle model also provide diminishing returns. The paper performs a number of additional experiments to identify the cause of the diminishing returns (accuracy of value estimates, variance of value estimates, magnitude & variance of actor/critic gradients) but concludes that none of the variables analyzed can account for the effect.

While the reviewers initially had concerns about the clarity of the paper and some of the results (especially in relation to the bias/variance trade-off) they felt these concerns were generally well-addressed by the rebuttal. In the discussion, the remaining concerns were: (1) that the paper neither answers the question of why the diminishing returns occur, nor provides further hypotheses; (2) that the messaging is unclear regarding what the diminishing returns are with respect to (i.e., that it's really with respect to model accuracy rather than horizon length); and (3) that the results are somewhat specific (to the particular class of continuous control problems in the function approximation setting, and to model-based value expansion methods rather than MBRL more broadly). However, it was generally agreed that despite these limitations, the paper studied an important question and that the insights would be useful to the MBRL community to know about. Therefore, I recommend acceptance.

In the discussion, we had some suggestions for how the authors can improve the paper for the final revision, which I would strongly recommend they do. (1) The paper should be more precise in its messaging that the diminishing returns are with respect to model accuracy, not horizon. Indeed, we agreed that it was not so surprising that longer horizons have minimal impact on data efficiency, as at some point the Q estimates will be sufficiently good---however, it is more surprising that the higher quality model has so little effect. (2) The paper should be more precise in stating that the effect studied is particular to one class of MBRL algorithms and might not exist when using other MBRL methods (e.g. Dyna, MPC, MuZero, etc.); in particular, we hypothesized that the diminishing returns would not happen with a perfect model in Dyna, because in Dyna data is simulated from the model, and if the model is perfect, then this is equivalent to simulating from the environment---so by definition, using a perfect model should always result in higher data efficiency. (3) Similarly, the paper would benefit from being clearer in the introduction that it focuses on a particular class of domains (continuous control in the function approximation setting) and the results may not generalize to other environments or the tabular setting.

We also had a hypothesis to explain the results, which the authors could consider. Specifically, the performance of the learning algorithms studied depends on two quantities: the data on which the losses are computed, and the Q-value estimate in the losses. In the algorithms studied, the model supports computing a more accurate Q-value, but it does not (directly) affect data collection. Perhaps the diminishing returns are due to the fact that a learned model already provides enough signal to compute a reasonably accurate Q-value using only a short horizon. However, the data used for learning is still just experience collected from the environment, and is only indirectly affected by the model-based value estimates (in the sense that those affect the behavior policy). So therefore, if the Q-values are already good enough, and the data distribution is (roughly) the same, we won't see much improvement. Perhaps if the model were also used to change the data distribution---e.g. via Dyna or MPC---then this could resolve the issue of diminishing returns.

**Note From Pc:**

if the above contains the word "oral" or "spotlight" please see: "oral" presentation means -> notable-top-5% and "spotlight" means -> notable-top-25%. As stated in our emails, we are disassociating presentation type from AC recommendations

**Summary Of Ac-Reviewer Meeting:**

While the reviewers initially had concerns about the clarity of the paper and some of the results (especially in relation to the bias/variance trade-off) they felt these concerns were generally well-addressed by the rebuttal. In the discussion, the remaining concerns were: (1) that the paper neither answers the question of why the diminishing returns occur, nor provides further hypotheses; (2) that the messaging is unclear regarding what the diminishing returns are with respect to (i.e., that it's really with respect to model accuracy rather than horizon length); and (3) that the results are somewhat specific (to the particular class of continuous control problems in the function approximation setting, and to model-based value expansion methods rather than MBRL more broadly). However, it was generally agreed that despite these limitations, the paper studied an important question and that the insights would be useful to the MBRL community to know about. The reviewers were ultimately supportive of accepting the paper, with some suggestions for how the authors could further improve the precision/clarity of the paper.